# Automated multiconformer model building for X-ray crystallography and cryo-EM

**Stephanie A Wankowicz[1]\*, Ashraya Ravikumar[1], Shivani Sharma[2,3], Blake Riley[2†], Akshay Raju[2], Daniel W Hogan[1], Jessica Flowers[1], Henry van den Bedem[1,4], Daniel A Keedy[2,5,6], James S Fraser[1]**

[1]Department of Bioengineering and Therapeutic Sciences, University of California, San Francisco, San Francisco, United States; [2]Structural Biology Initiative, CUNY Advanced Science Research Center, New York, United States; [3]Ph.D. Program in Biology, The Graduate Center, City University of New York, New York, United States; [4]Atomwise Inc, San Francisco, United States; [5]Department of Chemistry and Biochemistry, City College of New York, New York, United States; [6]Ph.D. Programs in Biochemistry, Biology and Chemistry, The Graduate Center, City University of New York, New York, United States

**\*For correspondence:**
mullane.stephanie@gmail.com

**Present address:** †Replay, San Diego, United States

**Abstract** In their folded state, biomolecules exchange between multiple conformational states that are crucial for their function. Traditional structural biology methods, such as X-ray crystallography and cryogenic electron microscopy (cryo-EM), produce density maps that are ensemble averages, reflecting molecules in various conformations. Yet, most models derived from these maps explicitly represent only a single conformation, overlooking the complexity of biomolecular structures. To accurately reflect the diversity of biomolecular forms, there is a pressing need to shift toward modeling structural ensembles that mirror the experimental data. However, the challenge of distinguishing signal from noise complicates manual efforts to create these models. In response, we introduce the latest enhancements to qFit, an automated computational strategy designed to incorporate protein conformational heterogeneity into models built into density maps. These algorithmic improvements in qFit are substantiated by superior $R_{free}$ and geometry metrics across a wide range of proteins. Importantly, unlike more complex multicopy ensemble models, the multiconformer models produced by qFit can be manually modified in most major model building software (e.g., Coot) and fit can be further improved by refinement using standard pipelines (e.g., Phenix, Refmac, Buster). By reducing the barrier of creating multiconformer models, qFit can foster the development of new hypotheses about the relationship between macromolecular conformational dynamics and function.

## eLife assessment

This article offers **important** updates to qFit, the state-of-the art tool for modeling alternative conformations of protein molecules based on high-resolution X-ray diffraction or cryo-EM data. While the authors provide **convincing** examples of qFit's performance, these are restricted to selected test cases. This article will be of interest to structural biologists and protein biochemists more generally.

## Introduction

Macromolecular X-ray crystallography and single-particle electron microscopy (cryo-EM) can provide valuable information on macromolecular conformational ensembles. These experiments cannot capture all conformations present in solution as many would disrupt the ability to obtain crystals

or align classifiable particles (*Cheng, 2015*). However, careful modeling from high-resolution X-ray crystallography and cryo-EM data can reveal widespread conformational heterogeneity, particularly for protein side chains and local backbone regions (*Smith et al., 1986*; *Herzik et al., 2017*). Such discrete, local conformational heterogeneity is significant for many biological functions, including macromolecular binding, catalysis, and allostery (*Keedy et al., 2018*; *Wankowicz et al., 2022*; *Yabu-karski et al., 2022*).

While the underlying data from X-ray diffraction and cryo-EM experiments contains information on temporal and spatial averages of tens of thousands to billions of protein copies, conventional structural modeling and refinement procedures fail to capture much of this valuable information. Most depositions in the Protein Data Bank reflect only an averaged, single ground state set of atomic coordinates (*Furnham et al., 2006*), ignoring weak but potentially biologically rich signals encoding alternative conformations sampled by distinct copies of the protein in the experiment.

Ideally, we would accurately model the complete ensemble of protein conformations reflected in experimental data (*Fraser et al., 2020*). The two ways to model the conformational heterogeneity present in the sample are to create ensembles or use alternative conformations (multiconformers) (*Woldeyes et al., 2014*). The PDB 'ensemble' format encodes multiple complete copies of the entire system in different models within a single file. Ensemble refinement approaches are implemented in phenix.ensemble_refinement (*Burnley et al., 2012*) and Vagabond (*Ginn, 2021*). In contrast, multi-conformers extend the conventional single-structure model by encoding each individual conformation using a distinct 'alternative location indicator (altloc)' within a single model. Altlocs are assigned distinct letters and can range from single atoms to a large number of connected or non-connected residues. Refinement and validation programs treat atoms sharing the same altloc as having the ability to interact with each other and with atoms lacking an altloc. In contrast, atoms with different altlocs cannot interact. By representing the underlying heterogeneity through discrete conformations with labeled altlocs, multiconformer models encode the distribution of states that contribute to the density map. Multiconformer models are notably easier to modify and more interpretable in software like Coot (*Emsley et al., 2010*), unlike ensemble methods that generate multiple complete protein copies (*Burnley et al., 2012*; *Ploscariu et al., 2021*; *Burling and Brünger, 1994*).

However, many factors make manually creating multiconformer models difficult and time-consuming. Interpreting weak density is complicated by noise arising from many sources, including crystal imperfections, radiation damage, and poor modeling (*Weichenberger et al., 2015*; *Kabsch, 2010*; *Karplus and Diederichs, 2012*) in X-ray crystallography, and errors in particle alignment and classification, poor modeling of beam-induced motion, and imperfect Detector Quantum Efficiency in high-resolution cryo-EM (*Glaeser, 2019*). These factors make visually distinguishing signals in Coot (*Emsley et al., 2010*) or other visualization software very difficult, especially when genuine low-occupancy signals overlap. Additionally, in X-ray crystallography, this process is iterative. Each time a new alternative conformation is placed, the resulting improvement in phases can impact the entire electron density map, often requiring adjustments to previously modeled regions. The difficulty of this process can lead to burnout and human bias, where parts of the protein are carefully modeled as multiconformers, whereas other regions remain modeled as single conformers. Despite these complications, multiconformer modeling can be implemented manually or using software such as FLEXR (*Stachowski and Fischer, 2023*) or qFit, as described below.

To enable more routine and impartial multiconformer modeling, we have previously developed qFit (*Keedy et al., 2015*; *Riley et al., 2021*; *van den Bedem et al., 2009*). This program leverages the ensemble-rich experimental data from density maps that are better than 2.0 Å resolution to automatically generate parsimonious multiconformer models (*Keedy et al., 2015*; *Riley et al., 2021*). As input, qFit takes a refined single-conformer structure and either a high-resolution X-ray or cryo-EM map as input, and then leverages powerful optimization algorithms to identify alternative protein (*Keedy et al., 2015*; *Riley et al., 2021*) or ligand (*van Zundert et al., 2018*) conformations.

Here, we present updates to qFit including algorithmic changes to protein conformation selection based on Bayesian information criteria (BIC), B-factor sampling, and updated cryo-EM scoring. Collectively, these advances enable the unsupervised generation of multiconformer models that routinely improve $R_{free}$ and model geometry metrics over single-conformer X-ray structures derived from high-resolution data across a diverse test set. We further demonstrate that qFit can identify alternative side-chain conformations in high-resolution cryo-EM datasets. With the improvements in

model quality outlined here, qFit can now increasingly be used for finalizing high-resolution models to derive ensemble-function insights.

## Results

### Overview of qFit protein algorithm

qFit protein is a tool that automatically identifies alternative conformations based on a high-resolution density map (generally better than ~2 Å) and a well-refined single-conformer structure (generally $R_{free}$ below 20%). For X-ray maps, we recommend using a composite omit map as input to minimize model bias (*Terwilliger et al., 2008*). For cryo-EM modeling applications, equivalent metrics of map and model quality are still developing, rendering the use of qFit for cryo-EM more exploratory.

Since our previous paper, we have made several modifications to the code, both algorithmically (e.g., scoring now includes BIC, and sampling of B-factors) and computationally (improving the efficiency and reliability of the code). All code and associated documentation can be found in the qFit GitHub repository (https://github.com/ExcitedStates/qfit-3.0, copy archived at *Wankowicz et al., 2024*). The version of qFit associated with this article is 2024.2 and is available at SBGrid (https://sbgrid.org/; *Morin et al., 2013*).

### qFit residue

For each residue, qFit samples backbone conformations, side-chain dihedral angles, and B-factors (*Figure 1A*). Using mixed quadratic programming (MIQP) and BIC, we select a parsimonious multi-conformer for each residue. The details of each component of this procedure are outlined below. The sampling and scoring of residues can be run in parallel using Python multiprocessing.

### Backbone sampling

The qFit process begins with sampling backbone conformations (*Figure 1A.1*). We first strip all hydrogens. For each residue, we perform a collective translation of backbone atom (N, C, Cα, O) coordinates. If the model has anisotropic B-factors, this translation is guided by the anisotropic B-factors of the Cβ. If anisotropic B-factors are absent, the translation of coordinates occurs in the Cα-Cβ, C-N, and (Cβ-Cα × C-N) directions. Each translation takes place in steps of 0.1 Å along each coordinate axis, extending to 0.3 Å, resulting in 9 (if isotropic) or 81 (if anisotropic) distinct backbone conformations for further analysis. For Gly and Ala, this is the only sampling that occurs.

### Aromatic angle sampling

For aromatic residues (His, Tyr, Phe, Trp), qFit takes the conformations from the backbone step (above) and builds part of the side chain out to Cγ (start of the aromatic ring) based on the input model coordinates (*Figure 1A.2*). Then, we alter the Cα-Cβ-Cγ angle ('the aromatic angle') in steps of ±3.75°, extending to ±7.5°, creating five partial side-chain conformations per backbone conformation. For non-aromatic residues, there is no sampling of this angle. These conformers provide variability in the placement of the aromatic ring prior to dihedral angle sampling.

### Dihedral angle sampling

The following steps occur for each $\chi$ dihedral angle for every residue (*Figure 1A.3*). For the first dihedral angle ($\chi 1$), the input is the sampled backbone conformations (or for aromatic residues the backbone and 'aromatic angle' conformers described above). We sample around the $\chi 1$ dihedral angle by enumerating a conformation every 6° for 24° on each side of an idealized rotamer (*Xie et al., 2020*) angle. rotamer ± around each rotamer. For proline, we sample the exo and endo conformations of the pyrrolidine ring, by ± 24° in steps of 6°. We then eliminate conformations that clash with other parts of the same sampled conformation of heavy atoms (based on hard spheres) or are redundant (using an all-atom root-mean-square deviation [RMSD] threshold of 0.01 Å).

These sampled conformations are then subjected to a quadratic programming (QP) optimization (*Agrawal et al., 2018*), which identifies the set of conformations whose weighted calculated density best fits the experimental electron density. The output of QP typically yields 5–15 conformations that best explain the density.

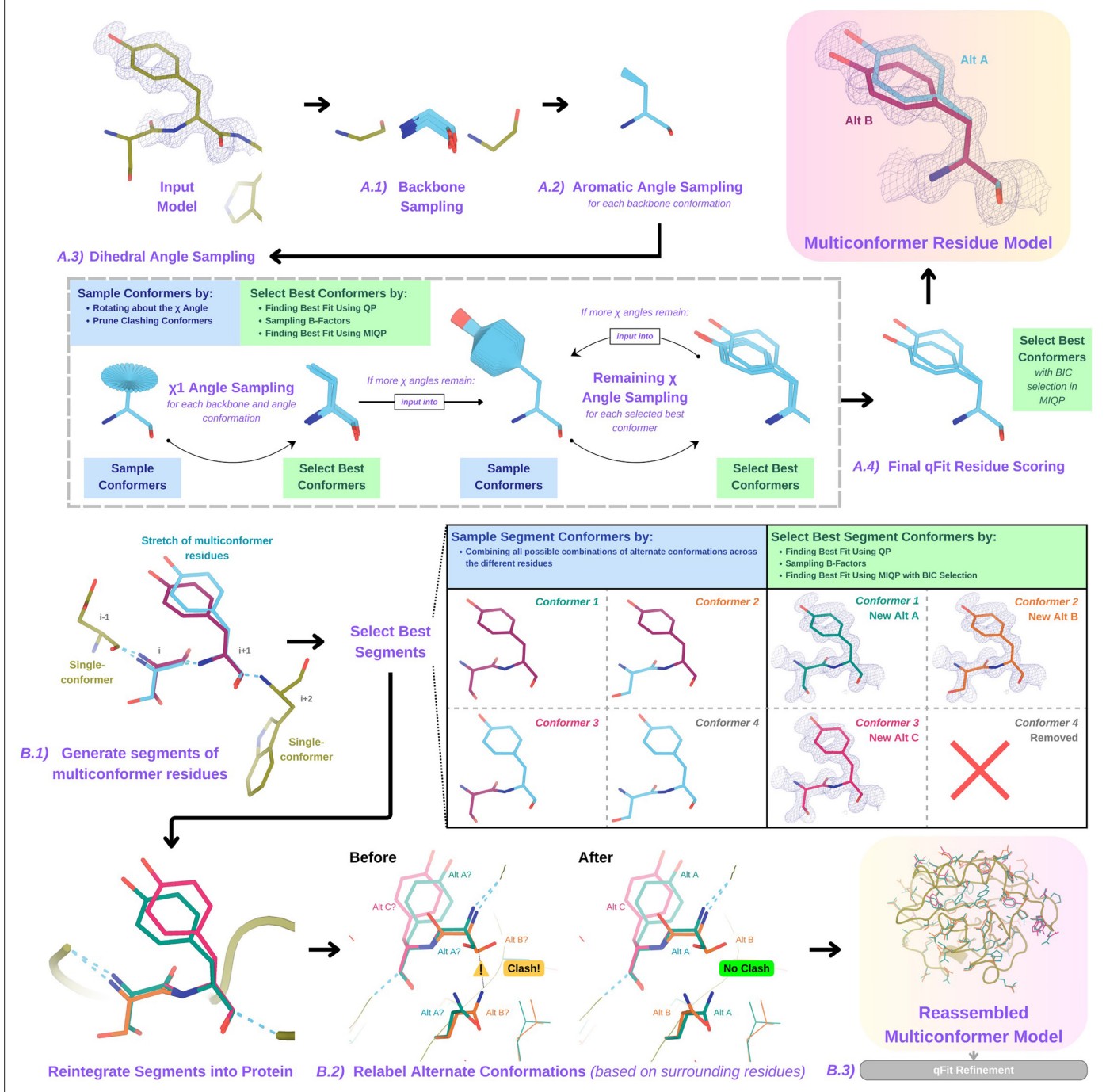

**Figure 1.** Programmatic flow of qFit protein algorithm. (**A**) qFit residue algorithm, demonstrated by Tyr118 in the E46Q mutant structure of the photoactive yellow protein from *Halorhodospira halophila* (PDB: 1OTA) (***Anderson et al., 2004***). The 2mFo-DFc composite omit density map contoured at 1 σ is shown as a blue mesh. (**A.1**) *Backbone sampling:* for each residue, qFit performs a collective translation of backbone atom (N, C, Cα, O) coordinates. (**A.2**) *Aromatic angle sampling:* for aromatic residues (His, Tyr, Phe, Trp), qFit takes the conformations from the backbone step and samples the Cα-Cβ-Cγ angle. (**A.3**) *Dihedral angle sampling:* since Tyr has two $\chi$ angles, qFit starts by taking the output conformers from the aromatic angle sampling step and exhaustively samples the $\chi$1 angle, scoring the best conformations based on QP/B-factor/mixed-integer quadratic programming (MIQP) scoring. qFit then uses these best conformations as input to sample the remaining $\chi$ angles in the Tyr residue. Since the only angle left to be sampled is the $\chi$2 angle, qFit rotates about the terminal ring of the Tyr and then scores the conformations that best fit the density. (**A.4**) *Final qFit residue scoring:* once we reach the terminal ring (all sampling steps have occurred), we perform QP and B-factor sampling, followed by MIQP with Bayesian information criteria (BIC) selection. MIQP with BIC selection removes a redundant overlapping conformation, resulting in two distinct conformations of this Tyr residue. This model is then output as the residue multiconformer. (**B**) qFit segment algorithm, demonstrated by

*Figure 1 continued on next page*

*Figure 1 continued*

Tyr118 in PDB: 1OTA. After identifying all optimal conformations for each individual residue, qFit works to connect the protein back together. (**B.1**) *qFit segment:* moving linearly along the protein sequence, qFit identifies 'segments' of residues with multiple backbone conformations. Here, Ser117 (i) and Tyr118 (i + 1) have multiple backbone conformations. qFit segment enumerates each possible combination of alternate conformations between these two residues, creating four possible combinations. The optimal combination of conformations is then determined by the QP/MIQP scoring, leading to one combination being culled. (**B.2**) *qFit relabel:* qFit uses Monte Carlo optimization with a steric model to assign altloc labels to spatially coupled alternative conformers. In this example, Ser117 and the neighboring Gln32 initially have clashing altloc B conformers. However, relabeling swaps the A and B labels of Gln32 to relieve this clash. (**B.3**) *qFit refinement:* we then refine the occupancies, coordinates, and B-factors of the raw qFit output file to produce a final qFit model. qFit improves overall fit to data relative to deposited structures.

Next, qFit samples the B-factors of the conformers. The input atomic B-factors are multiplied by a factor ranging from 0.5 to 1.5 in increments of 0.2. The resulting 50–150 conformation/B-factor combinations are subjected to a mixed-integer quadratic programming (MIQP) optimization. The MIQP algorithm incorporates two additional constraints relative to QP: a cardinality term, which limits the maximum number of conformations to 5, and a threshold term, which stipulates that no individual conformation can have an occupancy weight below 0.2. In qFit, MIQP then outputs up to five conformations.

For residues with subsequent dihedral angles, the conformations selected by the MIQP procedure at the $\chi$ (n-1) angle serve as the starting conformers for sampling the $\chi$ (n) angle. For residues with only one dihedral angle (Ser, Cys, Thr, Val, Pro), we proceed directly to scoring $\chi$ 1.

## Final qFit residue scoring

Upon reaching the terminal dihedral angle, we perform the optimization steps outlined above (QP/MIQP), but instead of relying only on the optimization algorithm to decide on the number of conformations to output, we also consider the model complexity (*Figure 1A.4*). qFit runs the MIQP step five times with a cardinality term ranging from 1 to 5. Taking each output, we calculate the BIC. The BIC provides a numerical value of the tradeoff between the difference between the calculated and experimental density (residual sum of squares) and the number of parameters (k). The number of parameters (k) is defined by the following: number of conformers * number of atoms * 4 (representing the x, y, z coordinates and B-factor). A heuristic scaling factor of 0.95 accounts for the fact that the coordinate parameters are not independent due to chemical constraints between atoms during sampling.

$$BIC = n^* In(rss/n) + k^* In(n)^* \; scaling \; factor$$

$$k = number \; of \; conformers^* number \; of \; atoms^* 4$$

$$rss = residual \; sum \; of \; squares$$

$$n = number \; of \; voxels \; in \; density \; map$$

$$scaling \; factor = 0.95$$

qFit then outputs the set of conformations with the lowest BIC value, concluding the qFit residue routine.

## Connecting residues together into a multiconformer model

After the sampling and scoring of each individual residue, qFit considers the entire protein together. First, we use MIQP and BIC to select the best-fitting conformations among connected residues, ensuring that neighboring backbone conformations have the same occupancy. Second, we label the alternative conformers while being aware of clashes.

### qFit segment

After identifying the optimal conformations for each residue in parallel, qFit reconnects the backbone atoms (*Figure 1B.1*). Moving from N- to C-terminus along the protein, we identify 'segments' of residues with multiple backbone conformations, delimited on each end by a residue with a single backbone conformation. The main reason for this step is to find a harmonious set of occupancies for adjacent residues in a segment. Within each segment, qFit creates fragments of three residues, enumerating all possible combinations of conformations in those residues, and selects the final combination of

conformations and their relative occupancies using the optimization algorithms outlined above. The BIC is modified for qFit segment such that k equals the number of conformations. qFit then moves along the protein, enumerating and selecting optimal combinations of fragment conformations until reaching the end of the segment.

## qFit relabel

Next, qFit determines the correct altloc labeling (A, B, C, D, E) of coupled alternative conformers using Monte Carlo optimization with a simple steric model of heavy atoms to prevent spatially adjacent conformers from sterically clashing (*Figure 1B.2*). There is also an option ('qFit segment only') to input a multiconformer model and run only the qFit segment and relabel procedures. This procedure can be especially helpful after manually adding or deleting conformations in Coot (*Emsley et al., 2010*). Running 'qFit segment only' will adjust the occupancy of the remaining conformations and correct the labeling of alternative conformations. This labeling step is not parallelized.

## qFit refinement

The raw output of qFit (a multiconfomer model) should then be refined. We provide scripts for a refinement procedure with Phenix (*Afonine et al., 2012*), where we iteratively refine the occupancy, coordinates, and B-factors, removing conformations with occupancies under 10%. Once the model is stable (has no conformations with occupancies less than 10%), we perform a final round of refinement which optimizes the placements of ordered water molecules ('Methods'). We then apply a mosaic bulk solvent (*phenix.mosaic*) to the final model, which allows for partial bulk solvent occupancy (*Afonine et al., 2024*). This refinement protocol outputs a final 'qFit model'. This model can then be examined and edited in Coot (*Emsley et al., 2010*) or other visualization software, and further refined using software such as Phenix.refine, Refmac, or Buster as the modeler sees fit.

To evaluate the impact of qFit algorithmic and code improvements, we collated a dataset of single-chain, unliganded, high-resolution (1.2–1.5 Å) protein X-ray crystallography structures from the PDB (*Berman et al., 2000*). We clustered these structures at a sequence identity threshold of 30% and selected the highest resolution structure per cluster. Finally, we ensured that the datasets ran without error through the qFit pipeline, including refinement with Phenix, resulting in 144 diverse structures (*Figure 2—figure supplement 1*).

Each deposited structure was initially re-refined using *phenix.refine* ('Methods') to eliminate differences from the original refinement protocols. The resulting re-refined model, which we refer to as the 'deposited model', was used as the input for qFit. Next, we ran qFit protein using the default parameters and refinement protocol to produce the 'qFit model'.

To evaluate the crystallographic modeling differences between the *deposited* and *qFit* models, we compared the $R_{free}$ values as an indicator of overall model/data agreement. The qFit model has a lower (improved) $R_{free}$ value for 76% (109/144) of structures (*Figure 2A*, *Figure 2—figure supplement 2A*, *Supplementary file 1*). On average, there is an absolute decrease of $R_{free}$ value by 0.6% (median deposited models $R_{free}$: 18.1%, median qFit models $R_{free}$: 17.5%), which is in line with theoretical expectations for the increase in model complexity created by qFit (*Holton et al., 2014*; *Vitkup et al., 2002*). $R_{free}$ is a valuable metric for monitoring overfitting, which is an important concern when increasing model parameters as is done in multiconformer modeling. An additional check on overfitting comes from monitoring R-gap, calculated as the difference between $R_{work}$ and $R_{free}$. qFit models have similar R-gap values compared to deposited models (mean: 3.0% for both models). Collectively, these results indicate that qFit improves the quality of most models without overfitting (*Figure 2—figure supplement 2B*).

Despite this general trend of improved models, 24% of the qFit models have worse $R_{free}$ than the deposited models (n = 35). The majority of these structures had a deposited model $R_{free}$ of over 20%. These high $R_{free}$ values are notable because our re-refinement procedure generally improved $R_{free}$ relative to the originally deposited model, particularly for structures with higher starting $R_{free}$ (*Figure 2—figure supplement 2C*). Since qFit builds off of the input structure and the map quality relies on model phases, accurately detecting alternative conformers depends heavily on the agreement between input model and data. This trend reinforced the idea that poor modeling in a deposited model, which serves as input to qFit, will result in poor performance of qFit. It further suggests

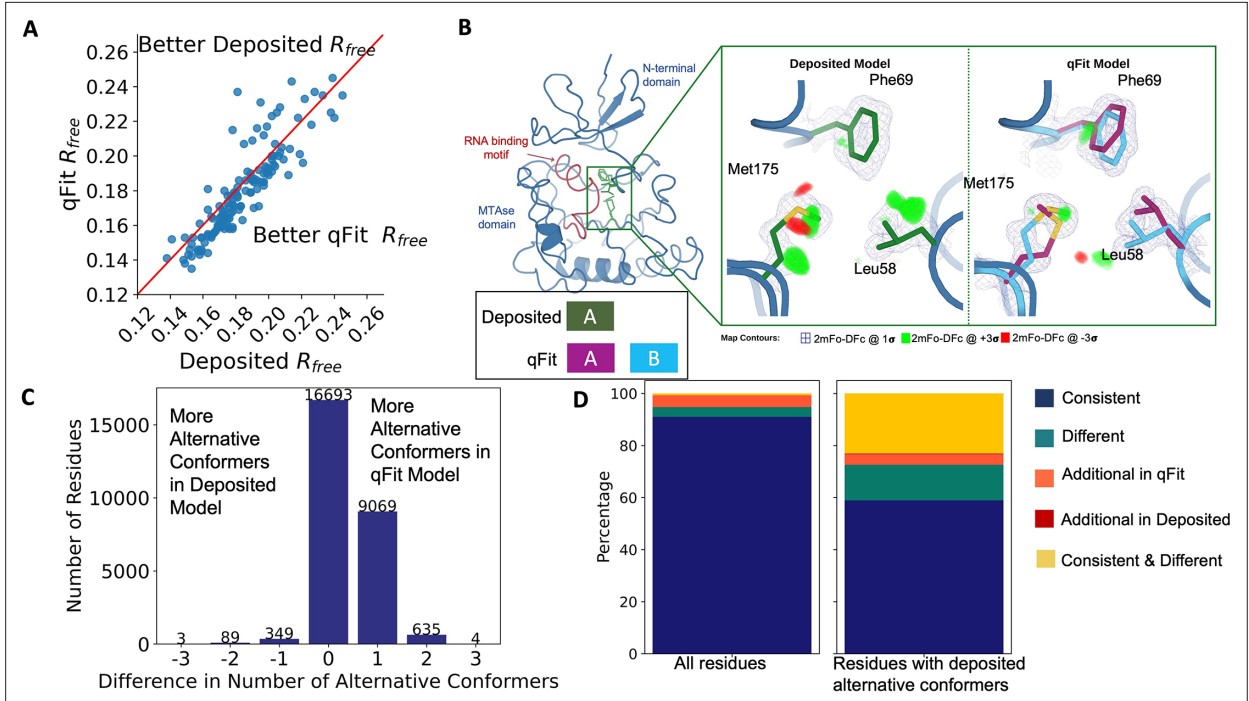

**Figure 2.** Multiconformer models created by qFit are better models than deposited single-conformer models. (**A**) The distribution of R$_{free}$ value in deposited models versus qFit models. The qFit R$_{free}$ values improve in 73% of structures.(**B**) qFit identifies new alternative conformations adjacent to the RNA binding motif in the *Pyrococcus horikoshii* fibrillarin pre-rRNA processing protein (PDB: 1G8A). (Left) qFit multiconformer model with the region in the right panel highlighted in green and the adjacent RNA binding motif highlighted in red. Key domains in the fibrillarin protein are also annotated in blue. (Right) Comparison of the deposited versus qFit model in a region with several conformationally heterogeneous residues. qFit identified new rotamers for Leu58 (tp) and Met175 (ttp and mtp) (*Lovell et al., 2000*) and significantly different alternative conformations within the original rotameric well for Phe69. (**C**) The differences in the number of alternative conformations per residue in deposited models versus qFit models. qFit adds at least one additional alternative conformation in 31.7% of residues (n = 9998). (**D**) The distribution of rotamer assignment agreement between the deposited and qFit models for different (sub)sets of residues. (Left) All residues (n = 42,626). (Right) Only residues with alternative conformations in the deposited model (n = 970). See main text for definitions of categories.

The online version of this article includes the following figure supplement(s) for figure 2:

**Figure supplement 1.** Flow diagram of the selection of the test set PDBs.

**Figure supplement 2.** R$_{free}$ and R-gap distributions.

**Figure supplement 3.** Examples of rotamer state categories.

that qFit is best employed at a late stage of modeling, after the single-structure model is of sufficient quality that it would be deposited in the PDB.

As an example of how qFit can uncover previously unnoticed conformational heterogeneity, we examined differences in conformations in the deposited versus qFit models of the *Pyrococcus horikoshii* fibrillarin pre-rRNA processing protein (PDB: 1G8A) (*Rodriguez-Corona et al., 2015*). We focused on the residues adjacent to the RNA binding motif. Among these residues, qFit identified well-justified alternative conformations for residues Leu58, Phe69, and Met175, including new rotamers for Leu58 and Met175, that were not present in the deposited model (*Figure 2B*). Beyond detecting alternative conformers in each of these residues, the qFit labeling process identified potential coupled motions between the alternative conformers. For example, when Leu58 is in the 'up' position (altloc A), Phe69 is also in the 'up' position (altloc A). It is possible that this coupled motion plays a role in RNA binding, a hypothesis that may merit further investigation.

## qFit recovers alternative conformations of deposited models and discovers new ones

As qFit mainly alters structures by adding alternative conformations, we examined the differences in the number of alternative conformations between the deposited models and qFit models. Only

2.9% of residues in the deposited models were multiconformers (two or more alternative conformations, n = 970). In contrast, 40.7% (n = 11,049) of residues in the qFit models were multiconformers (*Figure 2C*). The vast majority (92.5%) of multiconformer residues in the qFit models have only two alternative conformations; only 2.4% of residues have more than two alternative conformations.

Alternative conformations come in a few varieties. First and most obvious are alternative conformations that represent drastic changes in coordinates, most commonly in the form of rotameric changes. Most alternative conformations found in deposited models fall into this category. Second are more subtle changes in side-chain and backbone coordinates to represent heterogeneity within a rotameric state. This behavior is exemplified by the Tyr residue in *Figure 1A*. Third is even more subtle changes in coordinates to avoid strain because of the alternative conformations of neighboring residues (*Phenix, 2023*). This category is essentially imperceptible to visual inspection as the atom centers are nearly superimposable, but is important to avoid outlier bond geometry because of adjacent residues having larger displacements.

To quantify how often qFit models new rotameric states, we analyzed the qFit models with *phenix.rotalyze*, which outputs the rotamer state for each conformer (Methods; *Orengo et al., 1997*; *Lovell et al., 2000*). We classified the agreement between the deposited and qFit models into five categories (*Figure 2D*, *Figure 2—figure supplement 3*). The first category contains residues that have the same rotameric state(s) in both models. This category entails most single-conformer and multiconformer residues with agreement between the two models. Moreover, residues that have multiple conformations in the same rotamer in the qFit model (for the reasons described above) generally populated the same rotamer as found in single-conformer residues in the deposited models. Overall this category, 'Consistent', represents 93.7% of residues (n = 42,626) in the dataset.

The second and third categories deal with imbalance in alternative conformations that populate distinct rotamers. Since the original premise of qFit was to discover unmodeled alternative conformations, it is unsurprising that many residues in qFit models populate additional rotameric states that are absent in the deposited model. This category, *Additional Rotamer(s) in qFit model*, represents 2.38% of residues (n = 1082). In contrast, only two residues (0.06% of the dataset) are classified in the converse category, *Additional Rotamer(s) in deposited model*.

The final two categories cover disagreements in rotamer assignments. There are many cases where we observe only partial agreement between alternative conformers modeled in both the deposited and qFit models. These multiconformer residues share at least one common rotamer, but also populate alternative rotamers that are distinct between the two models. This behavior generally occurs in longer residues where subtle differences at higher $\chi$ angles leads to distinct rotameric assignments. This category, *Consistent & Different Rotamers*, represents 0.82% of residues (n = 373). The final category, *Different*, covers both multiconformer and single-conformer residues where there are no shared rotamer states between the two models. One reason this category occurs is for similar reasons as the *Consistent & Different* category: differences in terminal $\chi$ angles in weak density lead to distinct rotamer assignments. Another contributor to this category is single conformers, generally in the deposited model, modeled into density that qFit interprets as multiconformer. Often the rotamer modeled by the single conformer fits an 'average' rather than the two distinct minima fit by the multiconformer model. *Different* rotamer assignments represent 3.04% of residues (n = 1384). While the analyses above include all residues, focusing on residues that were modeled in as multiconformers in the deposited models (n = 970) reveals a large increase in the *Different* and *Consistent & Different Rotamers* categories, to 14.88% (n = 144) and 27.68% (n = 268) of residues, respectively. This increase highlights the sensitivity of the rotamer assignments and motivates benchmarking qFit on 'true positive' synthetic data in addition to deposited multiconformers.

Collectively, these analyses revealed that qFit identifies the majority of deposited alternative conformations and discovers new ones. Discrepancies between manually modeled and qFit alternative conformations predominantly result from weak density at terminal $\chi$ angles. When considered with the improvements in R$_{free}$, these results indicate that qFit is detecting more of the true underlying conformational heterogeneity that exists in crystallographic data.

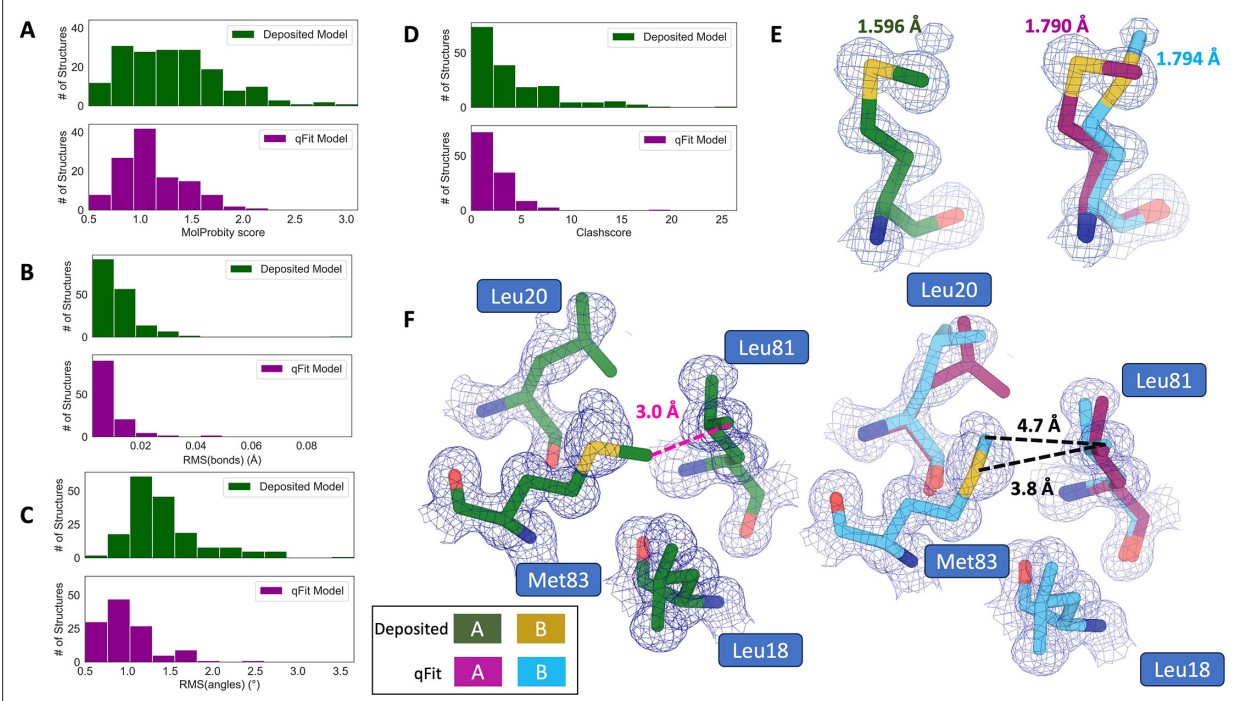

Figure 3. qFit improves some geometry metrics compared to deposited structures. (**A**) Model MolProbity score (deposited model: 1.27 (median) [0.94–0.16] (interquartile range), qFit model: 1.09 (median) [0.90–1.30] (interquartile range)), p-value = 0.006 from two-sided *t*-test. (**B**) Model averaged root-mean-square deviation (RMSD) (Å) of idealized versus model bond lengths (deposited model: 0.010 [0.0070–0.015], qFit model: 0.0073 [0.005–0.011]), p-value = 0.002 from two-sided *t*-test. (**C**) Model averaged RMSD (Å) of idealized versus model bond angles (deposited model: 1.30 [1.14–1.57], qFit model: 0.91 [0.77–1.13]), p-value = 3.79e-16 from two-sided *t*-test. (**D**). Model clashscore (deposited model: 2.50 [1.30–5.92], qFit model: 1.80 [1.31–3.73]), p-value = 0.0028 from two-sided *t*-test. (**E**). Example of qFit (right, blue, and magenta) fixing bond length by appropriately modeling in a second conformation. Meshes represent 2Fo-Fc density at 1 σ. Met189 from deposited structure (PDB: 1VF8; left, green) has a Sδ-Cε bond length of 1.596 Å (7.8 σ from idealized length of 1.791 Å) (*Williams et al., 2018*). qFit models two alternative conformations, filling in unmodeled density, and fixing the Sδ-Cε bond length (1.790 Å for alternative conformation A and 1.794 Å for alternative conformation B). (**F**) Example of qFit (right, blue, and magenta) fixing a clash between Met83 and Leu81 from deposited structure (PDB: 6HEQ). Meshes represent density at 1 σ. In the deposited model (left, green), Met83 is not correctly fitted into density and is clashing with Leu81 (closest contact: 3.0 Å). qFit corrects this by improving the fit of Met83, leading to the closest contact being 3.8 Å.

The online version of this article includes the following figure supplement(s) for figure 3:

**Figure supplement 1.** Deposited versus qFit model geometry.

## qFit improves multiple side-chain model geometry metrics

Although qFit improves the agreement of model to data by the addition of alternative conformations, we questioned whether this improvement comes at the cost of degrading model geometry. On one hand, the absence of geometric constraints in qFit backbone residue sampling and the connections made during qFit segment may result in worse geometry. On the other hand, placing additional alternative conformers may alleviate strain in the model that can result from fitting a single conformer into density that should be supported by multiple conformers (*Ginn, 2021*; *Stachowski and Fischer, 2023*; *Phenix, 2023*).

To validate geometry, we used MolProbity to evaluate the deposited and qFit models. MolProbity compares input models with idealized values and then provides component scores for various geometric and steric features that are summarized in an overall 'MolProbity score' (*Williams et al., 2018*). Component scores that examine all atoms (bond angle/length, clashscore) or side-chain atoms (rotamers) account for all alternative conformers. In contrast, scores that evaluate the backbone (Ramachandran, Cβ deviations) are reported for single-conformer residues or using only altloc A for multiconformer residues. Therefore, the overall MolProbity score includes some of the contributions of alternative conformations, but also misses the potential impact on some other aspects. In the future, we aim to explore updated metrics that consider all alternative conformations.

Compared to deposited models, qFit models had improved MolProbity scores (1.27 median deposited vs. 1.09 median qFit, p=0.006 from two-sided *t*-test; *Figure 3A*), which indicated that overall qFit improves the geometry while also usually improving fit to data. To further understand which parts of the model geometry were different (if any) between the deposited and qFit models, we explored the individual component scores and observed multiple component scores that improved in the qFit models. This included considerable improvements in bond lengths and angles in the qFit models (RMSD between idealized values for bond lengths: 0.010 Å median deposited vs. 0.007 Å median qFit, p=0.021 from two-sided *t*-test; RMSD between idealized values for bond angles: 1.30° median deposited vs. 0.91° median qFit, p=3.79e-16 from two-sided *t*-test; *Figure 3B and C*). We suspect that the primary factor behind this improvement was the incorporation of multiconformers, rather than straining a single conformer, to explain the density. To visualize an example of these differences, we investigated Met189 from PDB: 1V8F. In the deposited model, this residue has Sδ-Cε bond lengths of 1.596 Å, which are significantly shorter than the idealized lengths of 1.791 ± 0.025 Å (*Williams et al., 2018*). qFit adds an additional conformation, both explaining previously unmodeled density and bringing the Sδ-Cε bond lengths much closer to the expected values: 1.790 Å (alternative conformer A) and 1.794 Å (alternative conformer B) for the two conformations (*Figure 3E*). This multi-conformer residue with improved geometry is consistent with the hypothesis that qFit is alleviating strained geometry by modeling multiple conformations.

Additionally, qFit models have improved clashscores (2.50 median deposited, 1.80 median qFit, p=0.0028 from two-sided *t*-test; *Figure 3D*). We hypothesized that this was due to a mixture of modeling of alternative conformers and improved fit of single-conformer residues which are re-sampled and refined during the qFit procedure. We looked at the qFit modeling differences in a cluster of Met and Leu residues in PDB: 6HEQ, which had one of the largest changes in clashscores between the deposited and qFit models. We observed that qFit fixes the positioning of Met83, preventing the clash with both conformers of Leu81 and improving the local fit to density (*Figure 3F*).

We observed almost equivalent rotamer scores, favored Ramachandran values, and C-beta values (median number of rotamer outliers: 0.94 deposited vs. 0.800 qFit; percentage of Ramachandran favored: 97.7% deposited vs. 97.8% qFit; median value of clashscore: 2.50 deposited vs. 1.78 qFit) (*Figure 3—figure supplement 1*). Overall, the MolProbity scores suggest that qFit improved the model geometry, aligning with improved model/data agreement.

## Simulated data demonstrates qFit is appropriate for high-resolution data

In the previous sections, we established that qFit has the potential to improve R$_{free}$ and some geometry metrics relative to deposited structures. However, the vast majority of the residues in these deposited structures are modeled exclusively as single conformers. This homogeneity in single-conformation models limited our ability to assess how well qFit can recapitulate existing alternative conformers across a wide resolution range. To address this question, we generated artificial structure factors using an ultra-high-resolution structure (0.77 Å) of the SARS-CoV-2 Nsp3 macrodomain (PDB: 7KR0) (*Schuller et al., 2021*). This model had a high proportion of residues (47%) manually modeled as alternative conformations and did not employ qFit during model building or refinement, making it an ideal comparison structure. We refer to this structure as the 'ground truth 7KR0 model' and evaluated how well its alternative conformations were recapitulated by qFit as resolution was artificially worsened across synthetic datasets.

To create the dataset for resolution dependence, we used the ground truth 7KR0 model, including all alternative conformations, and generated artificial structure factors with a high-resolution limit ranging from 0.8 to 3.0 Å (in increments of 0.1 Å). We then added random noise to the structure factors that increased as resolution worsened ('Methods'; *Figure 4—figure supplement 1A, B*). To create a single-conformer model appropriate for input to qFit, we removed all alternative conformations from the ground truth model, maintaining all single conformations and altloc A. Next, we refined this single-conformer model against the synthetic datasets. Finally, we used the refined single-conformer model as input for qFit.

We then turned to evaluate the fidelity of qFit in recapitulating the ground truth 7KR0 model. For each residue, we first classified the residue as being a multiconformer or single conformer. Due to many residues in both the ground truth and qFit models having alternative conformations that nearly

overlap each other, we categorize residues as multiconformer only if they possess at least two alternative conformers with a side-chain heavy-atom RMSD greater than 0.5 Å. From this cutoff, 50 out of the 169 residues (30%) in the ground truth model are classified as multiconformers.

Next, we define each residue as having an agreement between the outputted qFit model and the ground truth 7KR0 model. If all qFit modeled conformers are within 0.5 Å of the deposited 7KR0 model, we classify it as a match. If not, we classify it as no match. A 'multiconformer match' has agreement between multiconformers across ground truth and qFit models; a 'single conformer match' has agreement between single conformers in the ground truth and qFit models. Generally, a 'multiconformer no match' has extra or distinct conformations in the qFit model; a 'single conformer no match' has at least one alternative conformation in the ground truth model that is not present in the qFit model or discordant single-conformer conformations.

We observed that qFit is consistently strong at capturing single-conformer residues (single conformer match) across resolutions. We did observe a drop off of detecting alternative conformations (multiconformer match) beyond resolutions of ~1.8–2.0 Å (*Figure 4B*, *Figure 4—figure supplement 1C*). This behavior is exemplified by Glu114, which is multiconformer in the ground truth model (*Figure 4C*). At high resolution (1.0 Å), qFit correctly models the alternative conformation and this residue is categorized as a multiconformer match. However, as resolution gets worse, qFit begins to mismodel this residue. At 1.8 Å resolution, qFit still models two alternative conformations and has a good fit to density; however, the secondary conformer has an RMSD greater than 0.5 Å away from the ground truth model; consequently, this residue is now categorized as a multiconformer no match. Finally, at 2.8 Å resolution, qFit only models a single conformer, moving the residue to the single conformer no match category.

## Simulated multiconformer data illustrate the convergence of qFit

Next, we tested the ability of qFit to detect alternative conformations over a larger, more diverse dataset. We generated artificial structure factors for the qFit models with improved $R_{free}$ values over the deposited values from the previous sections (n = 109). Although this dataset is more diverse, it has a notable weakness relative to the 7KR0 dataset test: the 7KR0 alternative conformations were modeled manually, whereas the larger dataset has alternative conformations modeled by qFit. Therefore, this second synthetic dataset assesses convergence of the qFit models across resolution.

Using these qFit models as ground truth models, we generated structure factors, performed refinement of single-conformer models, and ran qFit over the resolution range of 1.0–3.0 Å (*Figure 4—figure supplement 1A*). We observed a similar fall-off of multiconformer match residues around 2.0 Å (*Figure 4—figure supplement 1D*). Importantly, this dataset indicates that qFit still models single conformers well at lower resolutions. We also observe a trend of increased no match multiconformers/single conformers for longer residues that are just outside the 0.5 Å RMSD cutoff (*Figure 4—figure supplement 2*). We did not observe a relationship between input model $R_{free}$ and the number of correctly modeled conformers, but it is difficult to tell whether our synthetic noise procedures properly capture the dependence of qFit performance on input model/data agreement (*Figure 4—figure supplement 3A and B*).

We then assessed the agreement between individual conformers and the map. To do this, we used the Q-score (*Pintilie et al., 2020*), which compares the map profile of an atom with an ideal Gaussian distribution that would be observed if the atom perfectly fits into the density. Across the test dataset, residues that qFit models as single conformers have an almost equivalent Q-score to the ground truth model even at lower resolutions (*Figure 4D*). The primary alternative conformations in qFit models (occupancy between 0.5 and 1.0) and lower-occupancy alternative conformations (occupancy < 0.5) display Q-scores that are very close to the equivalent 'ground truth model' alternative conformations until a resolution of about 1.8 Å. At lower resolutions, there is a dramatic fall-off in model/map agreement for these alternative conformers. These trends were also observed with the 7KR0 dataset (*Figure 4—figure supplement 3C*). Overall, these analyses on both the 7KR0 and larger synthetic datasets confirm that qFit will best detect alternative conformations with high-resolution (1.8–2.0 Å or better) data.

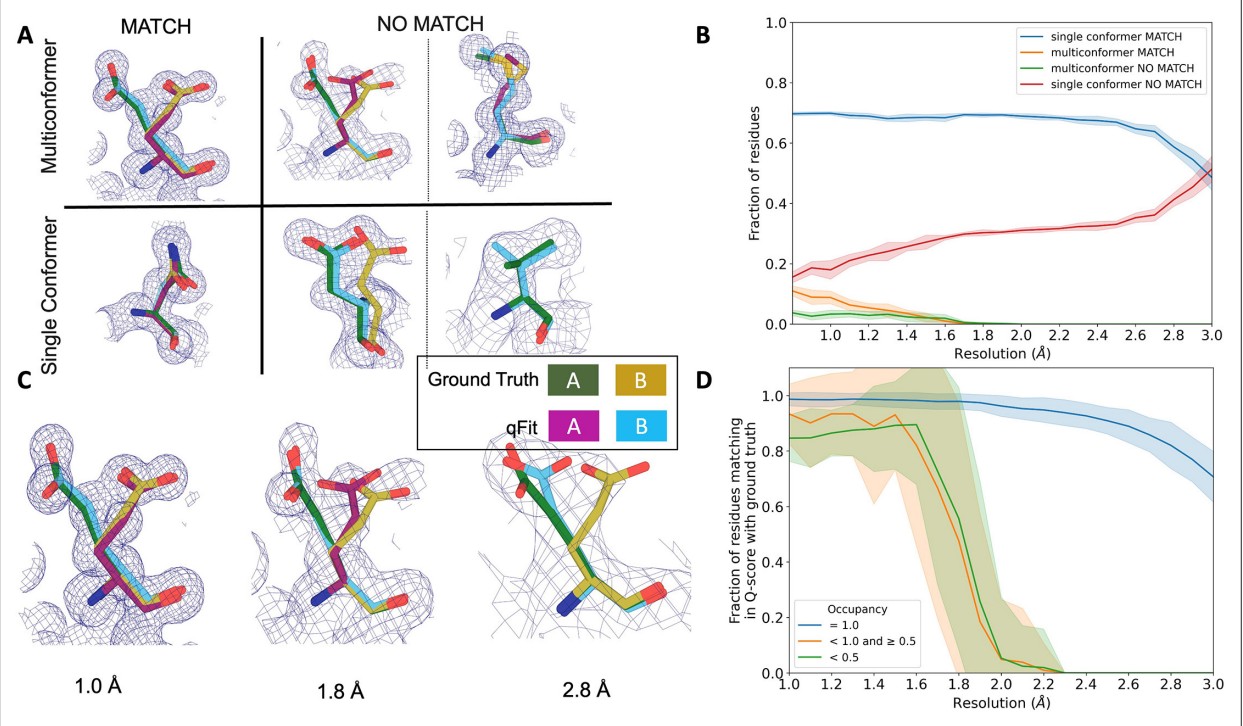

**Figure 4.** qFit performs best at high resolution of input dataset. (**A**) Ground truth model residues are shown as green and yellow sticks; qFit model residues are shown as magenta, cyan, and gray. Meshes represent density at 1 σ. Multiconformer match: residue is multiconformer in qFit model with root-mean-square deviation (RMSD) < 0.5 Å from ground truth residue. qFit models two distinct alternate conformations which recapitulate the ground truth residue's alternate conformations. Multiconformer no match: residue is multiconformer in qFit model with RMSD > 0.5 Å from ground truth residue. The example on the left has two alternate conformations in the ground truth. qFit models only one of them correctly. The example on the right is a single-conformation residue in ground truth but qFit models three alternate conformations. Single conformer match: residue is single-conformer in qFit model with RMSD < 0.5 Å from ground truth residue. Both ground truth model and qFit model have one distinct conformation and they align well. Single conformer no match: residue is single conformer in qFit model with RMSD > 0.5 Å from ground truth residue. The example on the left has two alternative conformations in the ground truth residue but only one conformation in the qFit residue. In the example on the right, the single conformer modeled by qFit does not align with the ground truth single conformer. (**B**) Proportion of all residues in the qFit models of 7KR0 that are modeled as multiconformer match (orange), single conformer match (blue), multiconformer no match (green), and single conformer no match (red) as a function of resolution of input synthetic data from the 7KR0 dataset. The shaded region denotes the 95% confidence interval. (**C**) Glu114 in the 7KR0 dataset modeled by qFit (cyan and magenta) compared to the ground truth structure (green and yellow) at different synthetic resolutions. Meshes represent density at 1 σ. (**D**) The fraction of residues in the qFit models of the qFit test dataset with a Q-score within 0.01 to that of the ground truth model as a function of resolution. In multiconformer residues, Q-score for every alternative conformation is calculated separately. Q-scores of residues (or) conformers which have matching occupancy (range) are compared. Occupancies of conformers were binned into three classes: occupancy equal to 1 (blue), 1 > occupancy ≥ 0.5 (orange) and occupancy < 0.5 (green).

The online version of this article includes the following figure supplement(s) for figure 4:

**Figure supplement 1.** Synthetic dataset generation and validation.

**Figure supplement 2.** Synthetic dataset statistics breakdown.

**Figure supplement 3.** Comparison of $R_{free}$ statistic and occupancy across synthetic dataset.

## qFit models alternative conformers in cryo-EM density maps

As single-particle cryo-EM is increasingly producing high-resolution (better than 2 Å) reconstructions where alternative conformers can be detected (*Nakane et al., 2020*; *Xie et al., 2020*), we wanted to improve and test the ability of qFit to model alternative conformations guided by cryo-EM maps. While a previous version of qFit introduced cryo-EM compatibility (*Riley et al., 2021*), we had not optimized the approach to work with cryo-EM maps and models. qFit can now be run in 'EM mode' which uses electron structure factors, improves the treatment of solvent background levels, and reduces the default maximum number of alternative conformations (cardinality) ('Methods').

To benchmark our ability to model alternative conformations in high-resolution cryo-EM structures, we initially gathered a dataset of 22 structures with a depositor-provided resolution better than 2 Å

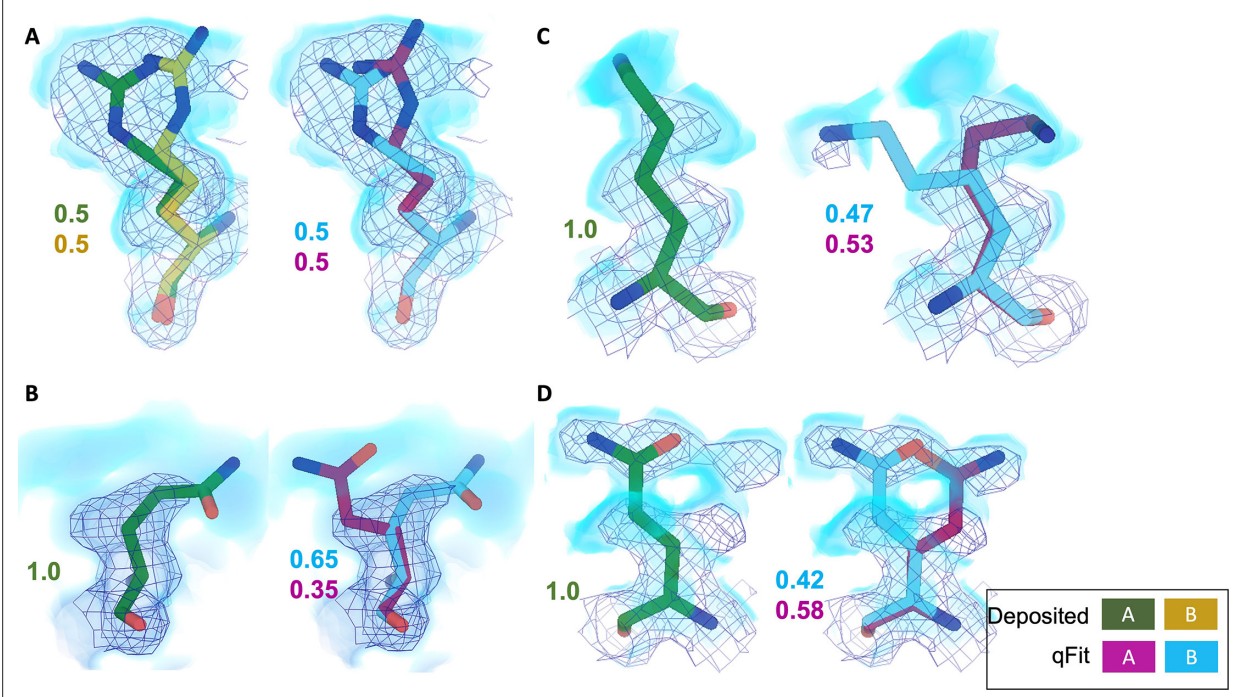

**Figure 5.** qFit identifies alternative conformations in high-resolution cryo-EM models. Meshes represent density at 1 σ, with blue volumes representing density at 0.5 σ. Green and yellow sticks represent deposited conformation(s). Cyan and magenta sticks represent qFit conformations. Occupancy is labeled based on each conformer. (**A**) qFit recapitulated the deposited alternative conformations of Arg22 (chain A) in apoferritin (PDB: 7A4M, resolution: 1.22 Å). (**B**) qFit identified a previously unmodeled alternative conformation of Glu14 (chain A) in apoferritin (PDB: 7A4M, resolution: 1.22 Å). (**C**) qFit identified a previously unmodeled alternative conformation of Lys49 (chain A) in a different structure of apoferritin (PDB: 6Z9E, resolution: 1.55 Å). (**D**) qFit identified a previously unmodeled alternative conformation of Gln403 (chain A) in adeno-associated virus (PDB: 7KFR, resolution: 1.56 Å).

The online version of this article includes the following figure supplement(s) for figure 5:

**Figure supplement 1.** qFit improes some geometry metrics in cryo-EM models.

(Fourier shell correlation [FSC] at 0.143). However, only eight of these structures have a resolution better than 2 Å (FSC at 0.143) when calculated by the Electron Microscopy Data Bank (EMDB) (*Chiu et al., 2021*). Some of the original 22 structures did not have FSC curves in EMDB (n = 6) due to a lack of data, and others had an EMDB calculated resolution worse than 2 Å (n = 8) (*Supplementary file 2*). The absence of standardized maps for determining cryo-EM structure resolution complicated our selection of structures for qFit analysis.

We downloaded the eight models with resolution better than 2 Å from the PDB and their corresponding maps from EMDB. Using the default parameters of *phenix.autosharpen*, we sharpened all maps and re-refined each structure (*phenix.real_space_refine*) against its sharpened map. qFit was run with the 'EM' flag and the output model was refined using the qFit real space refinement script ('Methods').

Across the first asymmetric unit of the eight models, 8.21% (n = 64) of residues in the deposited model had at least two alternative conformers in the deposited structure compared with 39.6% (n = 266) in the qFit model. To determine whether qFit could recapitulate the modeling of alternative conformers from deposited structures, we compared the high-resolution apoferritin deposited model (PDB: 7A4M, resolution: 1.22 Å) with the qFit model using the same criteria outlined in the resolution dependence section above (RMSD within 0.5 Å). qFit correctly models 77% of residues in the first asymmetric unit. This includes Arg22, which has two alternative conformations in the deposited model. qFit was able to recapitulate both alternative conformations (*Figure 5A*), highlighting that qFit can detect manually modeled alternative conformations in cryo-EM maps. In addition, qFit detected several unmodeled alternative conformers that were visually confirmed (*Figure 5B–D*).

As with the X-ray models, we wanted to determine how qFit changes the model geometry. Similar to the X-ray models, we observed that qFit improves bond lengths and angles, and similar Cβ deviations. Unlike the observations in the X-ray dataset, qFit does increase (worsen) the MolProbity score, likely coming from high clashscore of most structures, highlighting a future improvement in the algorithm (*Figure 5—figure supplement 1*).

While we have made significant progress in modeling alternative conformations in cryo-EM data, the lack of consistent map handling, validation, and metrics with cryo-EM structures and maps is a major impediment to further development. Even among this select group of structures, there were varying levels of experimental and computational map details on EMDB and in manuscripts (*Nakane et al., 2020*; *Xie et al., 2020*; *Yip et al., 2020*), including information on masking, handling of bulk solvent, and local resolution. Our approach depends on sampling and scoring based on resolution. While there is an accepted formula for calculating resolution (FSC at 0.143), the maps to calculate these are not consistent, leading to differences in resolution as we observed between the deposited versus EMDB calculated resolutions. Further, resolution can vary across a single model, and metrics for such local resolutions are not always widely available. Additionally, the handling of background bulk solvent values varies widely, from masking to flattening these values. New methods for cryo-EM ensemble modeling will benefit from ongoing efforts to standardize the storage of raw, meta, and processed data (*Kleywegt et al., 2024*).

## Discussion

Structural biology plays a vital role in understanding the complex connection between protein structure and function. However, since proteins exist as ensembles, structural biology modeling approaches need to adapt accordingly. X-ray crystallography and cryo-EM data hold significant information on these ensembles that is often ignored. qFit offers a solution by leveraging powerful optimization algorithms to transform well-modeled single-conformer models into multiconformer models. Here we demonstrate that qFit can uncover widespread conformational heterogeneity that better represents the true underlying conformational ensemble data as demonstrated by lower $R_{free}$ values. Further, we determine that qFit can reliably pick up on alternative conformers that were modeled manually, highlighting that qFit could be used as a tool to significantly speed up modeling of high-resolution structures.

This automation in modeling is needed especially in light of advances in data collection automation and fast detectors. These tools have revolutionized the field of X-ray crystallography, enabling high-temperature datasets, time-resolved experiments, and high-throughput data collection (*Wolff et al., 2022*; *Dasgupta et al., 2019*; *Correy et al., 2022*; *Mehlman et al., 2022*; *Ebrahim et al., 2021*). With the ability to capture different conformations, there is a growing demand for methods that can detect protein alternative conformers to extract as much biological information as possible. This is highlighted in massive ligand-soaking campaigns (*Schuller et al., 2021*; *Gahbauer et al., 2023*; *Douangamath et al., 2020*; *Günther et al., 2021*), where there are often hundreds of structures with different ligands to parse. qFit provides a key tool to help extract the most out of these structures by improving the models and providing a better jumping-off point to determine how ligand binding impacts the protein. However, our data here show that not only does qFit need a high-resolution map to be able to detect signal from noise, it also requires a very well-modeled structure as input.

While both throughput and resolution are currently lower for cryo-EM, recent high-resolution maps have observable conformational heterogeneity (*Nakane et al., 2020*; *Yip et al., 2020*). Current classification approaches do not allow sorting based on signals as small as alternative side-chain conformations (*Zhong et al., 2021*; *Chen and Ludtke, 2021*; *Kinman et al., 2023*), necessitating approaches like qFit for modeling. We see great potential in combining qFit with classification approaches to understand conformational heterogeneity at different scales. In the future, qFit can likely be applied more widely to EM maps in regions with high local resolution (*Terashi et al., 2022*). In addition, we will also incorporate modeling of nucleic acids, with an emphasis on automating refinement of alternative base positions in high-resolution ribosome structures in future work (*Li et al., 2020*; *Fromm et al., 2023*; *Hintze et al., 2017*). However, we encountered many difficulties in applying qFit to EM data relative to the more established X-ray data. In particular, there are still disparities in how maps are sharpened and how masks are used to exclude noise or lower experimental signals, such as solvent (*Wang et al., 2022*; *Lawson et al., 2021*), making it very challenging to evaluate whether models,

especially multiconformer or ensemble models, have improved fit to the data. We suggest strengthening guidelines for reporting computational processing and improving validation tools to gauge agreement between models and cryo-EM maps (*Wang et al., 2022*; *Lawson et al., 2021*; *Burley et al., 2022*).

We envision many other future improvements that will further enhance the quality and accuracy of multiconformer models for both X-ray crystallography and cryo-EM. Simulations have demonstrated that subpar modeling of the macromolecule(s) and surrounding solvent is a major potential avenue to further reduce R-factors (*Holton et al., 2014*; *Vitkup et al., 2002*). To accurately account for water molecules in multiconformer models, partially occupied water molecules must be identified and labeled in connection with protein atoms. Automated detection and refinement of partial-occupancy waters should help improve fit to experimental data (*Weichenberger et al., 2015*) and provide additional insights into hydrogen-bond patterns and the influence of solvent on alternative conformations (*Weichenberger et al., 2015*).

Additionally, while qFit models have overall improved geometry in some respects relative to single-conformer models, we still have room for improvement for fixing backbone metrics (Ramachandran and Cβ deviations). The geometry improvements are likely mostly due to single-conformer models having strained conformations that fit the 'mean' conformation rather than multiple partially overlapping conformations. Further gains in both accuracy and geometry quality will emerge with better sampling of backbone conformations (*Keedy et al., 2015*). Such improvements are important because splitting the backbone, where appropriate, can result in detection of biologically important side-chain alternative conformations (*Davis et al., 2006*). Notably, the recently described FLEXR approach, which leverages Ringer and Coot to model alternative side chains into density peaks, illustrates that many gains can be made with side-chain focused modeling alone (*Stachowski and Fischer, 2023*). However, further improvements to backbone modeling, including larger-scale motions such as alternative loop conformations (*Biel et al., 2017*) or coordinated larger-scale shifts of secondary-structural elements (*Deis et al., 2014*; *Fraser et al., 2011*), will likely yield even higher-quality multiconformer models.

Lastly, experimental and computational advancements in structural biology have increased the focus on ensemble-based models (*Ginn, 2021*; *Riley et al., 2021*; *Chen and Ludtke, 2021*; *Kinman et al., 2023*; *Pearce and Gros, 2021*). But the current data format for structural models (PDB, mmCIF) does not allow for more complex representation of ensembles. qFit is compatible with manual modification and further refinement as long as the subsequent software uses the PDB standard altloc column, as is common in most popular modeling and refinement programs. The models can therefore generally also be deposited in the PDB using the standard deposition and validation process. However, to even more appropriately capture the many aspects of ensembles, we would ideally like to have multiple nested ensembles representing both larger and local conformational changes, or to be able to show how two different backbone conformations can each be 'parents' to different side-chain conformations (*Wankowicz and Fraser, 2024*). Currently, neither the PDB nor CIF format allows for this type of representation (*Hancock et al., 2022*; *Pearce et al., 2017*; *Vallat et al., 2023*).

In summary, qFit drastically reduces the time and effort required to create multiconformer models from X-ray and cryo-EM data, thereby lowering the barrier to generating new hypotheses about the relationship between conformational ensembles and biological function (*Keedy et al., 2018*; *Wankowicz et al., 2022*; *Otten et al., 2018*; *Zaragoza et al., 2023*). Additionally, qFit can provide key data to bridge to the next frontier of structure prediction. While AlphaFold (*Jumper et al., 2021*) has achieved stunning success in predicting protein structure by training against single-conformation models, future improvements to structure prediction might be gained by more accurately modeling the extent of conformational heterogeneity (*Lane, 2023*).

## Methods
### Generating and running the qFit test set

To test the impact of algorithmic changes in qFit, we created a dataset of 144 high-resolution (1.2–1.5 Å) X-ray crystallography structures deposited in the PDB (*Supplementary file 1*). These were single-chain protein structures (in the asymmetric unit and at the level of biological assembly) and contained no ligands or mutations. The maximum sequence identity between any two structures was

set as 30%. Based on CATH classification (*Orengo et al., 1997*), the resultant entries represented 72 folds (*Supplementary file 1*). The structures represented 24 space groups. All these structures were re-refined as described in 'Initial refinement protocol'. These re-refined models are referred to as deposited models. To create multiconformer models, we input the re-refined structures in qFit protein, followed by the post qFit refinement protocol. These multiconformer models are referred to as qFit models.

### Initial refinement protocol

All structures from the PDB were re-refined using *phenix.refine* with the following parameters:

> *refinement.refine.strategy=\*individual_sites \*individual_adp \*occupancies*
> *refinement.output.serial=5*
> *refinement.main.number_of_macro_cycles=5*
> *refinement.main.nqh_flips = False*
> *refinement.output.write_maps = False*
> *refinement.hydrogens.refine=riding*
> *refinement.main.ordered_solvent = True*
> *refinement.target_weights.optimize_xyz_weight = true*
> *refinement.target_weights.optimize_adp_weight = true*

The re-refined models were used as the input for subsequent qFit models.

### Running qFit

For this analysis, qFit was run using the following command from qFit version 2023.1.

### X-ray

*qfit_protein composite_omit_map.mtz -l 2FOFCWT,PH2FOFCWT rerefine_pdb.pdb.*

### Cryo-EM

*qfit_protein sharpened_map.ccp4 rerefine_cryo-EM.pdb -r <resolution> -em -n 10 -s 5.*

### qFit new features

#### Parallelization of large maps

Often, cryo-EM maps are very large and reach memory limits using Python multiprocessing. Multiprocessing is used to model multiple residues independently in parallel. We have now implemented a new scheme to divide the density map into portions centered around each residue of interest and feed those portions of the map into our parallelization.

#### B-factor sampling

To sample B-factors along with atomic coordinates at each step of qFit residue, we first perform one round of quadratic programming to reduce the number of conformations. For all remaining conformations, the input B-factor of each atom in the residue is multiplied by 0.5–1.5 in increments of 0.2. All conformations with sampled B-factors and coordinates are inputs for MIQP.

#### Bayesian information criteria

BIC was implemented in the final selection of residue and segment conformations. BIC is defined as the real space residual correlation coefficient penalized by the number of parameters (k):

> *BIC = n \* np.log(rss / n)+k \* np.log(n) \* 0.95*
> *rss = residual sum of squares*
> *n = number of datapoints*
> In qFit residue, k is defined as
> *k = 4 \* number of atoms \* number of conformations*
> In qFit segment, k is defined as
> *k = number of conformations*

BIC is calculated for each candidate cardinality (1–5). We then choose the set of conformations with the lowest BIC as the final conformations for the residue or segment under consideration.

## Iterative optimization algorithm with non-convex problems

Due to our exhaustive sampling, there are times when the MIQP optimization algorithm fails to find a non-convex solution. To address this limitation, we have implemented a procedure that iteratively removes solutions one-by-one based on the two solutions with the closest RMSD until MIQP identifies a solution.

## Implementation of open-source QP/MIQP algorithms

qFit previously relied on IBM CPLEX to score conformations. While this is free to academics, it is not open source. We have switched to CVXPY, an open-source QP and MIQP solver (*Agrawal et al., 2018*; *Diamond and Boyd, 2016*).

## Occupancy constraints

To help refine segments (i.e., sets of residues with alternative conformations flanked by residues with only a single conformation) during X-ray refinement, we now output a restraint file at the end of the qFit protein run for X-ray refinement. This restraint file enables 'group occupancy refinement' for residues in a segment with the same alternative conformation. In group occupancy refinement, all residues within the group are refined to the same occupancy, reducing the free parameters to fit.

## Finalizing qFit models with iterative refinement

We iteratively run five macrocycles of refinement followed by a script that removes any conformations with occupancy less than 0.10. This script also renormalizes the occupancies of any remaining conformations in that segment, ensuring that the occupancy sums to 1. This procedure ends when no conformations have a refined occupancy of less than 0.10 or after 50 total rounds of refinement (whichever comes first). Afterward, we perform one final refinement where we release the occupancy constraints on the segments, turn on automated solvent picking, and optimize B-factors (specified as ADP parameters in Phenix) and coordinate weights.

## Cryo-EM

To improve the detection of alternative conformations in cryo-EM structures, we made some key updates to part of the qFit algorithm. All of these updates to the algorithm will turn on with the *-em* flag. First, we now use electron scattering factors when calculating the modeled electron density. Second, we have removed bulk solvent electron density values (set at 0.3 in X-ray qFit protein). We also restricted the occupancy threshold cardinality to be 0.3 (compared to 0.2 in X-ray qFit protein) to reduce misplaced conformations.

## Q-score

We implemented the option for users to use Q-scores to determine whether qFit should be run on a residue or not. This option is off by default. To utilize this feature, first generate Q-scores by using the *mapq.py* script, which is included in the Q-score command-line interface package (https://github.com/gregdp/mapq, copy archived at *gregdp, 2023*). qFit takes in a text file of Q-scores by using the *–qscore* option in *qFit_protein*. By default, all residues with a Q-score of less than 0.7 are not modeled as multiconformers, but are considered in qFit segment. Users can also adjust this level by using the *–qscore_cutoff* option in qFit protein.

## qFit-segment-only runs

qFit can be used as a tool along with iterative model building and refinement. If a user manually removes or adds additional conformations using Coot (*Emsley et al., 2010*) or similar software, this can disrupt the occupancy sum of the residue and the connectivity of the backbone. To alleviate such problems, we developed an option (*qfit_protein –only-segment*) to facilitate manual model adjustment after running qFit. This procedure generates connected backbones with consistent occupancies for coupled neighboring conformers.

For example, suppose residue n has four alternative backbone conformations (A, B, C, D) and residue n+1 has two alternative conformations (A, B). In that case, this procedure will create C and D conformers for residue n+1 by duplicating its A and B conformers. This duplication continues until we reach the end of a segment so that all backbones have the same number of alternative conformations (A, B, C, D) and are, therefore, properly connected. Subsequent crystallographic refinement of this model (see 'Post-qFit refinement script' above) will cause the duplicated conformations to diverge slightly and will behave as expected without introducing geometry errors.

## Analysis metrics

Scripts for all metrics can be found in the scripts folder in the qFit GitHub repository (https://github.com/ExcitedStates/qfit-3.0, copy archived at *Wankowicz et al., 2024*). Our scripts for running qFit protein on an SGE-based server and all scripts for figures can be found at https://github.com/fraser-lab/qFit_biological_testset/tree/main (copy archived at *Wankowicz and Ravikumar, 2024*).

### R-values

R-values were obtained after the final round of refinement for the re-refined deposited models (*deposited_rerefine.sh*) and for the qFit models after the iterative refinementscript (*qfit_final_xray_refine.sh*).

### B-factors

For each residue, we calculate an occupancy weighted B-factor (each heavy atom B-factor is weighted by its occupancy). For each heavy atom, we calculate the weighted using the following formula:

$$Occupancy\ Weighted\ B\text{-}factor = Occupancy * (4*pi/B\text{-}factor)^{1.5}$$

### Rotamers

The rotamer name for each alternative conformation was determined by *phenix.rotalyze* (*Williams et al., 2018*) while manually relaxing the outlier criteria to 0.1%. Rotamers were compared on a residue-by-residue basis. To compare rotamers, we only consider the first two $\chi$ dihedral angles. Each residue was classified into four categories: same, additional rotamer in qFit model, additional rotamer in the deposited model, or different.

## Generating synthetic data for resolution dependence

To generate artificial electron density data at increasingly poorer resolutions, we first increased the B-factors of all atoms of the ground truth model by 1 $Å^2$ for every 0.1 Å reduction in resolution and placed the models in a P1 box. We randomly shook the coordinates using the *shake* argument in *phenix.pdbtools* with root-mean-square error of shaking given as 0.2 * desired resolution of synthetic data. We generated structure factors ($F_{shake}$) for each of these shaken models from 0.8 Å to 3.0 Å in increments of 0.1 Å using the *phenix.fmodel* command-line function (with bulk solvent parameters k_sol = 0.4, b_sol = 45, and 5% R-free flags). We then added noise to the structure factors as follows:

$$F_{noisy} = F_{shake} + (sqrt(F_{shake}) * random\ number\ from\ normal\ distribution * resolution\ of\ model * 0.5)$$

The scaling factors of 0.2 and 0.5 for shake RMSD and noise addition were determined by trying out different values and identifying the values which gave the lowest $R_{free}$ over the resolution range after refining the model against the generated structure factors. The addition of noise to $F_{shake}$ was done using the *sftools* command in CCP4 (*Winn et al., 2011*). Then, the ground truth model with adjusted B-factors was stripped of alternative conformations (if any) at every residue position. The resulting single-conformer model was refined with the $F_{noisy}$ structure factors (*Figure 4—figure supplement 1*).

The final refined model was given as input to qFit and the composite omit map was obtained for the $F_{noisy}$ structure factors. The multiconformer model given by qFit was refined with *phenix.refine* as explained in the post-qFit refinement script section. Since there is some randomness involved in simulating noise in the synthetic datasets, at each resolution, we generate 10 synthetic datasets and apply the qFit protocol to each one. The same steps of data synthesis were followed for the larger qFit test dataset containing 103 models, except that one set of structure factors was generated for each model at each resolution instead of 10 as in the 7KR0 dataset.

## Match classifications for synthetic data

Match multiconformer residues were those with at least two alternative conformations and an RMSD of less than 0.5 Å between the ground truth and qFit model conformations (e.g., qFit model altloc A has an RMSD of less than 0.5 Å to ground truth model altloc A or B, and qFit model altloc B has an RMSD of less than 0.5 Å to the other ground truth model altloc A or B) (*Figure 4A*). No match multiconformer residues have at least two alternative conformations in the qFit model, but fewer conformations in the ground truth model (*Figure 4A*). Alternatively, for a no match multiconformer residue, if the ground truth model residue is also multiconformer, then the RMSD between at least one of the conformations of qFit residue and ground truth residue is more than 0.5 Å (*Figure 4A*). A match single conformer residue is when both the ground truth and qFit model have a single conformer and they have an RMSD of less than 0.5 Å (*Figure 4A*). A no match single conformer residue is when the qFit model has a single conformer but the ground truth model has more than one alternative conformer or both models have a single conformer but they have an RMSD greater than 0.5 Å (*Figure 4A*).

## Acknowledgements

This work was supported by a National Institutes of Health (NIH) grant GM145238 and 1125 Chan Zuckerberg Initiative Essential Open Software grant to JSF and NIH R35 1126 GM133769 to DAK. We thank Christopher Williams and Vincent Chen for help with 1127 interpretations of MolProbity score ideal side-chain geometry.

## Additional information

### Competing interests

Henry van den Bedem: The work in this publication does not overlap with Henry van den Bedem's role at Atomwise Inc, and there is no conflict of interest. The other authors declare that no competing interests exist.

### Funding

| Funder | Grant reference number | Author |
|---|---|---|
| National Institutes of Health | GM145238 | James S Fraser |
| National Institutes of Health | GM133769 | Daniel A Keedy |
| Chan Zuckerberg Initiative | EOSS5 | James S Fraser |

The funders had no role in study design, data collection and interpretation, or the decision to submit the work for publication.

### Author contributions

Stephanie A Wankowicz, Conceptualization, Software, Formal analysis, Supervision, Visualization, Methodology, Writing – original draft, Project administration, Writing – review and editing, Data curation; Ashraya Ravikumar, Data curation, Formal analysis, Visualization, Writing – original draft, Writing – review and editing; Shivani Sharma, Formal analysis; Blake Riley, Software, Methodology, Writing – review and editing; Akshay Raju, Formal analysis, Visualization, Writing – review and editing; Daniel W Hogan, Software, Methodology; Jessica Flowers, Software; Henry van den Bedem, Conceptualization, Methodology, Writing – review and editing; Daniel A Keedy, James S Fraser, Supervision, Funding acquisition, Writing – original draft, Writing – review and editing

### Author ORCIDs

Stephanie A Wankowicz ⓘ http://orcid.org/0000-0002-4225-7459
Daniel W Hogan ⓘ http://orcid.org/0000-0003-3375-408X
Daniel A Keedy ⓘ http://orcid.org/0000-0002-9184-7586
James S Fraser ⓘ http://orcid.org/0000-0002-5080-2859

Reviewer #1 (Public Review): https://doi.org/10.7554/eLife.90606.3.sa1
Reviewer #2 (Public Review): https://doi.org/10.7554/eLife.90606.3.sa2
Reviewer #3 (Public Review): https://doi.org/10.7554/eLife.90606.3.sa3
Author response https://doi.org/10.7554/eLife.10.7554/eLife.90606.3.3.sa4

## Additional files

### Supplementary files

• Supplementary file 1. High-resolution X-ray crystallography dataset information including PDB, R-free/R-work.

• Supplementary file 2. High-resolution cryo-EM dataset information including resolution, geometry information.

• MDAR checklist

### Data availability

All qFit models for the PDBs discussed in this paper are included in Zenodo deposition https://doi.org/10.5281/zenodo.10936292. Code can be found at https://github.com/ExcitedStates/qfit-3.0 (copy archived at *Wankowicz et al., 2024*).

The following dataset was generated:

| Author(s) | Year | Dataset title | Dataset URL | Database and Identifier |
|---|---|---|---|---|
| Wankowicz S | 2024 | Uncovering Protein Ensembles: Automated Multiconformer Model Building for X-ray Crystallography and Cryo-EM | https://doi.org/10.5281/zenodo.10936291 | Zenodo, 10.5281/zenodo.10936291 |

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
