## [Editor Report · eLife assessment]

This article offers **important** updates to qFit, the state-of-the art tool for modeling alternative conformations of protein molecules based on high-resolution X-ray diffraction or cryo-EM data. While the authors provide **convincing** examples of qFit's performance, these are restricted to selected test cases. This article will be of interest to structural biologists and protein biochemists more generally.

---

## [Referee Report · Reviewer #1 (Public Review)]

Summary:

Protein conformational changes are often critical to protein function, but obtaining structural information about conformational ensembles is a challenge. Over a number of years, the authors of the current manuscript have developed and improved an algorithm, qFit protein, that models multiple conformations into high resolution electron density maps in an automated way. The current manuscript describes the latest improvements to the program, and analyzes the performance of qFit protein in a number of test cases, including classical statistical metrics of data fit like Rfree and the gap between Rwork and Rfree, model geometry, and global and case-by-case assessment of qFit performance at different data resolution cutoffs. The authors have also updated qFit to handle cryo-EM datasets, although the analysis of its performance is more limited due to a limited number of high-resolution test cases and less standardization of deposited/processed data.

Strengths:

The strengths of the manuscript are the careful and extensive analysis of qFit's performance over a variety of metrics and a diversity of test cases, as well as the careful discussion of the limitations of qFit. This manuscript also serves as a very useful guide for users in evaluating if and when qFit should be applied during structural refinement.

---

## [Referee Report · Reviewer #2 (Public Review)]

Summary:

The manuscript by Wankowicz et al. describes updates to qFit, an algorithm for the characterization of conformational heterogeneity of protein molecules based on X-ray diffraction of Cryo-EM data. The work provides a clear description of the algorithm used by qFit. The authors then proceed to validate the performance of qFit by comparing it to deposited X-ray entries in the PDB in the 1.2-1.5 Å resolution range as quantified by Rfree, Rwork-Rfree, detailed examination of the conformations introduced by qFit, and performance on stereochemical measures (MolProbity scores). To examine the effect of experimental resolution of X-ray diffraction data, they start from an ultra high-resolution structure (SARS-CoV2 Nsp3 macrodomain) to determine how the loss of resolution (introduced artificially) degrades the ability of qFit to correctly infer the nature and presence of alternate conformations. The authors observe a gradual loss of ability to correctly infer alternate conformations as resolution degrades past 2 Å. The authors repeat this analysis for a larger set of entries in a more automated fashion and again observe that qFit works well for structures with resolutions better than 2 Å, with a rapid loss of accuracy at lower resolution. Finally, the authors examine the performance of qFit on cryo-EM data. Despite a few prominent examples, the authors find only a handful (8) of datasets for which they can confirm a resolution better than 2.0 Å. The performance of qFit on these maps is encouraging and will be of much interest because cryo-EM maps will, presumably, continue to improve and because of the rapid increase in the availability of such data for many supramolecular biological assemblies. As the authors note, practices in cryo-EM analysis are far from uniform, hampering the development and assessment of tools like qFit.

Strengths:

qFit improves the quality of refined structures at resolutions better than 2.0 A, in terms of reflecting true conformational heterogeneity and geometry. The algorithm is well designed and does not introduce spurious or unnecessary conformational heterogeneity. I was able to install and run the program without a problem within a computing cluster environment. The paper is well written and the validation thorough.

I found the section on cryo-EM particularly enlightening, both because it demonstrates the potential for discovery of conformational heterogeneity from such data by qFit, and because it clearly explains the hurdles towards this becoming common practice, including lack of uniformity in reporting resolution, and differences in map and solvent treatment.

Weaknesses:

The authors begin the results section by claiming that they made "substantial improvement" relative to the previous iteration of qFit, "both algorithmically (e.g., scoring is improved by BIC, sampling of B factors is now included) and computationally (improving the efficiency and reliability of the code)" (bottom of page 3). However, the paper does not provide a comparison to previous iterations of the software or quantitation of the effects of these specific improvements, such as whether scoring is improved by the BIC, how the application of BIC has changed since the previous paper, whether sampling of B factors helps, and whether the code faster. It would help the reader to understand what, if any, the significance of each of these improvements was.

The exclusion of structures containing ligands and multichain protein models in the validation of qFit was puzzling since both are very common in the PDB. This may convey the impression that qFit cannot handle such use cases. (Although it seems that qFit has an algorithm dedicated to modeling ligand heterogeneity and seems to be able to handle multiple chains). The paper would be more effective if it explained how a user of the software would handle scenarios with ligands and multiple chains, and why these would be excluded from analysis here.

It would be helpful to add some guidance on how/whether qFit models can be further refined afterwards in Coot, Phenix, ..., or whether these models are strictly intended as the terminal step in refinement.

Appraisal & Discussion:

Overall, the authors convincingly demonstrate that qFit provides a reliable means to detect and model conformational heterogeneity within high-resolution X-ray diffraction datasets and (based on a smaller sample) in cryo-EM density maps. This represents the state of the art in the field and will be of interest to any structural biologist or biochemist seeking to attain an understanding of the structural basis of the function of their system of interest, including potential allosteric mechanisms-an area where there are still few good solutions. That is, I expect qFit to find widespread use.

---

## [Referee Report · Reviewer #3 (Public Review)]

Summary:

The authors address a very important issue of going beyond a single-copy model obtained by the two principal experimental methods of structural biology, macromolecular crystallography and cryo electron microscopy (cryo-EM). Such multiconformer model is based on the fact that experimental data from both these methods represent a space- and time-average of a huge number of the molecules in a sample, or even in several samples, and that the respective distributions can be multimodal. Different from structure prediction methods, this approach is strongly based on high-resolution experimental information and requires validated single-copy high-quality models as input. Overall, the results support the authors' conclusions.

In fact, the method addresses two problems which could be considered separately:

- An automation of construction of multiple conformations when they can be identified visually;

- A determination of multiple conformations when their visual identification is difficult or impossible.

The first one is a known problem, when missing alternative conformations may cost a few percent in R-factors. While these conformations are relatively easy to detect and build manually, the current procedure may save significant time being quite efficient, as the test results show.

The second problem is important from the physical point of view and has been addressed first by Burling & Brunger (1994; https://doi.org/10.1002/ijch.199400022). The new procedure deals with a second-order variation in the R-factors, of about 1% or less, like placing riding hydrogen atoms, modeling density deformation or variation of the bulk solvent. In such situations, it is hard to justify model improvement. Keeping Rfree values or their marginal decreasing can be considered as a sign that the model is not overfitted data but hardly as a strong argument in favor of the model.

In general, overall targets are less appropriate for this kind of problem and local characteristics may be better indicators. Improvement of the model geometry is a good choice. Indeed, yet Cruickshank (1956; https://doi.org/10.1107/S0365110X56002059) showed that averaged density images may lead to a shortening of covalent bonds when interpreting such maps by a single model. However, a total absence of geometric outliers is not necessarily required for the structures solved at a high resolution where diffraction data should have more freedom to place the atoms where the experiments "see" them.

The key local characteristic for multi conformer models is a closeness of the model map to the experimental one. Actually, the procedure uses a kind of such measure, the Bayesian information criteria (BIC). Unfortunately, there is no information about how sharply it identifies the best model, how much it changes between the initial and final models; in overall there is not any feeling about its values. The Q-score (page 17) can be a tool for the first problem where the multiple conformations are clearly separated and not for the second problem where the contributions from neighboring conformations are merged. In addition to BIC or to even more conventional target functions such as LS or local map correlation, the extreme and mean values of the local difference maps may help to validate the models.

This method with its results is a strong argument for a need in experimental data and information they contain, differently from a pure structure prediction. At the same time, absence of strong density-based proofs may limit its impact.

Strengths:

Addressing an important problem and automatization of model construction for alternative conformations using high-resolution experimental data.

Weaknesses:

An insufficient validation of the models when no discrete alternative conformations are visible and essentially missing local real-space validation indicators.

---

## [Author Response]

The following is the authors’ response to the original reviews.

**Public Reviews:**

**Reviewer #1 (Public Review):**
Summary:Protein conformational changes are often critical to protein function, but obtaining structural information about conformational ensembles is a challenge. Over a number of years, the authors of the current manuscript have developed and improved an algorithm, qFit protein, that models multiple conformations into high resolution electron density maps in an automated way. The current manuscript describes the latest improvements to the program, and analyzes the performance of qFit protein in a number of test cases, including classical statistical metrics of data fit like Rfree and the gap between Rwork and Rfree, model geometry, and global and case-by-case assessment of qFit performance at different data resolution cutoffs. The authors have also updated qFit to handle cryo-EM datasets, although the analysis of its performance is more limited due to a limited number of high-resolution test cases and less standardization of deposited/processed data.Strengths:The strengths of the manuscript are the careful and extensive analysis of qFit's performance over a variety of metrics and a diversity of test cases, as well as the careful discussion of the limitations of qFit. This manuscript also serves as a very useful guide for users in evaluating if and when qFit should be applied during structural refinement.
**Reviewer #2 (Public Review):**
SummaryThe manuscript by Wankowicz et al. describes updates to qFit, an algorithm for the characterization of conformational heterogeneity of protein molecules based on X-ray diffraction of Cryo-EM data. The work provides a clear description of the algorithm used by qFit. The authors then proceed to validate the performance of qFit by comparing it to deposited X-ray entries in the PDB in the 1.2-1.5 Å resolution range as quantified by Rfree, Rwork-Rfree, detailed examination of the conformations introduced by qFit, and performance on stereochemical measures (MolProbity scores). To examine the effect of experimental resolution of X-ray diffraction data, they start from an ultra high-resolution structure (SARS-CoV2 Nsp3 macrodomain) to determine how the loss of resolution (introduced artificially) degrades the ability of qFit to correctly infer the nature and presence of alternate conformations. The authors observe a gradual loss of ability to correctly infer alternate conformations as resolution degrades past 2 Å. The authors repeat this analysis for a larger set of entries in a more automated fashion and again observe that qFit works well for structures with resolutions better than 2 Å, with a rapid loss of accuracy at lower resolution. Finally, the authors examine the performance of qFit on cryo-EM data. Despite a few prominent examples, the authors find only a handful (8) of datasets for which they can confirm a resolution better than 2.0 Å. The performance of qFit on these maps is encouraging and will be of much interest because cryo-EM maps will, presumably, continue to improve and because of the rapid increase in the availability of such data for many supramolecular biological assemblies. As the authors note, practices in cryo-EM analysis are far from uniform, hampering the development and assessment of tools like qFit.StrengthsqFit improves the quality of refined structures at resolutions better than 2.0 A, in terms of reflecting true conformational heterogeneity and geometry. The algorithm is well designed and does not introduce spurious or unnecessary conformational heterogeneity. I was able to install and run the program without a problem within a computing cluster environment. The paper is well written and the validation thorough.I found the section on cryo-EM particularly enlightening, both because it demonstrates the potential for discovery of conformational heterogeneity from such data by qFit, and because it clearly explains the hurdles towards this becoming common practice, including lack of uniformity in reporting resolution, and differences in map and solvent treatment.WeaknessesThe authors begin the results section by claiming that they made "substantial improvement" relative to the previous iteration of qFit, "both algorithmically (e.g., scoring is improved by BIC, sampling of B factors is now included) and computationally (improving the efficiency and reliability of the code)" (bottom of page 3). However, the paper does not provide a comparison to previous iterations of the software or quantitation of the effects of these specific improvements, such as whether scoring is improved by the BIC, how the application of BIC has changed since the previous paper, whether sampling of B factors helps, and whether the code faster. It would help the reader to understand what, if any, the significance of each of these improvements was.

Indeed, it is difficult (embarrassingly) to benchmark against our past work due to the dependencies on different python packages and the lack of software engineering. With the infrastructure we’ve laid down with this paper, made possible by an EOSS grant from CZI, that will not be a problem going forward. Not only is the code more reliable and standardized, but we have developed several scientific test sets that can be used as a basis for broad comparisons to judge whether improvements are substantial. We’ve also changed with “substantial improvement” to “several modifications” to indicate the lack of comparison to past versions.

The exclusion of structures containing ligands and multichain protein models in the validation of qFit was puzzling since both are very common in the PDB. This may convey the impression that qFit cannot handle such use cases. (Although it seems that qFit has an algorithm dedicated to modeling ligand heterogeneity and seems to be able to handle multiple chains). The paper would be more effective if it explained how a user of the software would handle scenarios with ligands and multiple chains, and why these would be excluded from analysis here.

qFit can indeed handle both. We left out multiple chains for simplicity in constructing a dataset enriched for small proteins while still covering diversity to speed the ability to rapidly iterate and test our approaches. Improvements to qFit ligand handling will be discussed in a forthcoming work as we face similar technical debt to what we saw in proteins and are undergoing a process of introducing “several modifications” that we hope will lead to “substantial improvement” - but at the very least will accelerate further development.

It would be helpful to add some guidance on how/whether qFit models can be further refined afterwards in Coot, Phenix, ..., or whether these models are strictly intended as the terminal step in refinement.

We added to the abstract:

“Importantly, unlike ensemble models, the multiconformer models produced by qFit can be manually modified in most major model building software (e.g. Coot) and fit can be further improved by refinement using standard pipelines (e.g. Phenix, Refmac, Buster).”

and introduction:

“Multiconformer models are notably easier to modify and more interpretable in software like Coot12 unlike ensemble methods that generate multiple complete protein copies (Burnley et al. 2012; Ploscariu et al. 2021; Temple Burling and Brünger 1994).”

and results:

“This model can then be examined and edited in Coot12 or other visualization software, and further refined using software such as phenix.refine, refmac, or buster as the modeler sees fit.”

and discussion

“qFit is compatible with manual modification and further refinement as long as the subsequent software uses the PDB standard altloc column, as is common in most popular modeling and refinement programs. The models can therefore generally also be deposited in the PDB using the standard deposition and validation process.”

Appraisal & DiscussionOverall, the authors convincingly demonstrate that qFit provides a reliable means to detect and model conformational heterogeneity within high-resolution X-ray diffraction datasets and (based on a smaller sample) in cryo-EM density maps. This represents the state of the art in the field and will be of interest to any structural biologist or biochemist seeking to attain an understanding of the structural basis of the function of their system of interest, including potential allosteric mechanisms-an area where there are still few good solutions. That is, I expect qFit to find widespread use.
**Reviewer #3 (Public Review):**
Summary:The authors address a very important issue of going beyond a single-copy model obtained by the two principal experimental methods of structural biology, macromolecular crystallography and cryo electron microscopy (cryo-EM). Such multiconformer model is based on the fact that experimental data from both these methods represent a space- and time-average of a huge number of the molecules in a sample, or even in several samples, and that the respective distributions can be multimodal. Different from structure prediction methods, this approach is strongly based on high-resolution experimental information and requires validated single-copy high-quality models as input. Overall, the results support the authors' conclusions.In fact, the method addresses two problems which could be considered separately:- An automation of construction of multiple conformations when they can be identified visually;

- A determination of multiple conformations when their visual identification is difficult or impossible.

We often think about this problem similarly to the reviewer. However, in building qFit, we do not want to separate these problems - but rather use the first category (obvious visual identification) to build an approach that can accomplish part of the second category (difficult to visualize) without building “impossible”/nonexistent conformations - *with a consistent approach/bias.*

The first one is a known problem, when missing alternative conformations may cost a few percent in R-factors. While these conformations are relatively easy to detect and build manually, the current procedure may save significant time being quite efficient, as the test results show.

We agree with the reviewers' assessment here. The “floor” in terms of impact is automating a tedious part of high resolution model building and improving model quality.

The second problem is important from the physical point of view and has been addressed first by Burling & Brunger (1994; https://doi.org/10.1002/ijch.199400022). The new procedure deals with a second-order variation in the R-factors, of about 1% or less, like placing riding hydrogen atoms, modeling density deformation or variation of the bulk solvent. In such situations, it is hard to justify model improvement. Keeping Rfree values or their marginal decreasing can be considered as a sign that the model is not overfitted data but hardly as a strong argument in favor of the model.

We agree with the overall sentiment of this comment. What is a significant variation in R-free is an important question that we have looked at previously (http://dx.doi.org/10.1101/448795) and others have suggested an R-sleep for further cross validation (https://pubmed.ncbi.nlm.nih.gov/17704561/). For these reasons it is important to get at the significance of the changes to model types from large and diverse test sets, as we have here and in other works, and from careful examination of the *biological* significance of alternative conformations with experiments designed to test their importance in mechanism.

In general, overall targets are less appropriate for this kind of problem and local characteristics may be better indicators. Improvement of the model geometry is a good choice. Indeed, yet Cruickshank (1956; https://doi.org/10.1107/S0365110X56002059) showed that averaged density images may lead to a shortening of covalent bonds when interpreting such maps by a single model. However, a total absence of geometric outliers is not necessarily required for the structures solved at a high resolution where diffraction data should have more freedom to place the atoms where the experiments "see" them.

Again, we agree—geometric outliers should not be completely absent, but it is comforting when they and model/experiment agreement both improve.

The key local characteristic for multi conformer models is a closeness of the model map to the experimental one. Actually, the procedure uses a kind of such measure, the Bayesian information criteria (BIC). Unfortunately, there is no information about how sharply it identifies the best model, how much it changes between the initial and final models; in overall there is not any feeling about its values. The Q-score (page 17) can be a tool for the first problem where the multiple conformations are clearly separated and not for the second problem where the contributions from neighboring conformations are merged. In addition to BIC or to even more conventional target functions such as LS or local map correlation, the extreme and mean values of the local difference maps may help to validate the models.

We agree with the reviewer that the problem of “best” model determination is poorly posed here. We have been thinking a lot about htis in the context of Bayesian methods (see: https://www.ncbi.nlm.nih.gov/pmc/articles/PMC9278553/); however, a major stumbling block is in how variable representations of alternative conformations (and compositions) are handled. The answers are more (but by no means simply) straightforward for ensemble representations where the entire system is constantly represented but with multiple copies.

This method with its results is a strong argument for a need in experimental data and information they contain, differently from a pure structure prediction. At the same time, absence of strong density-based proofs may limit its impact.

We agree - indeed we think it will be difficult to further improve structure prediction methods without much more interaction with the experimental data.

Strengths:Addressing an important problem and automatization of model construction for alternative conformations using high-resolution experimental data.Weaknesses:An insufficient validation of the models when no discrete alternative conformations are visible and essentially missing local real-space validation indicators.

While not perfect real space indicators, local real-space validation is implicit in the MIQP selection step and explicit when we do employ Q-score metrics.

**Recommendations for the authors:**

**Reviewer #1 (Recommendations For The Authors):**
A point of clarification: I don't understand why waters seem to be handled differently in for cryo-EM and crystallography datasets. I am interested about the statement on page 19 that the Molprobity Clashscore gets worse for cryo-EM datasets, primarily due to clashes with waters. But the qFit algorithm includes a round of refinement to optimize placement of ordered waters, and the clashscore improves for the qFit refinement in crystallography test cases. Why/how is this different for cryo-EM?

We agree that this was not an appropriate point. We believe that the high clash score is coming from side chains being incorrectly modeled. We have updated this in the manuscript and it will be a focus of future improvements.

**Reviewer #2 (Recommendations For The Authors):**
- It would be instructive to the reader to explain how qFit handles the chromophore in the PYP (1OTA) example. To this end, it would be helpful to include deposition of the multiconformer model of PYP. This might also be a suitable occasion for discussion of potential hurdles in the deposition of multiconformer models in the PDB (if any!). Such concerns may be real concerns causing hesitation among potential users.

Thank you for this comment. qFit does not alter the position or connectivity of any HETATM records (like the chromophore in this structure). Handling covalent modifications like this is an area of future development.

Regarding deposition, we have noted above that the discussion now includes:

“qFit is compatible with manual modification and further refinement as long as the subsequent software uses the PDB standard altloc column, as is common in most popular modeling and refinement programs. The models can therefore, generally also be deposited in the PDB using the standard deposition and validation process.”

Finally, we have placed all PDBs in a Zenodo deposition (XXX) and have included that language in the manuscript. It is currently under a separate data availability section (page XXX). We will defer to the editor as to the best header that should go under.

- It may be advisable to take the description of true/false pos/negatives out of the caption of Figure 4, and include it in a box or so, since these terms are important in the main text too, and the caption becomes very cluttered.

We think adding the description of true/false pos/negatives to the Figure panel would make it very cluttered and wordy. We would like to retain this description within the caption. We have also briefly described each in the main text.

- page 21, line 4: some issue with citation formatting.

We have updated these citations.

- page 25, second paragraph: cardinality is the number of members of a set. Perhaps "minimal occupancy" is more appropriate.

Thank you for pointing this out. This was a mistake and should have been called the occupancy threshold.

- page 26: it's - its

Thank you, we have made this change.

- Font sizes in Supplementary Figures 5-7 are too small to be readable.

We agree and will make this change.

**Reviewer #3 (Recommendations For The Authors):**
General remarks(1) As I understand, the procedure starts from shifting residues one by one (page 4; A.1). Then, geometry reconstruction (e.g., B1) may be difficult in some cases joining back the shifted residues. It seems that such backbone perturbation can be done more efficiently by shifting groups of residues ("potential coupled motions") as mentioned at the bottom of page 9. Did I miss its description?

We would describe the algorithm as sampling (which includes minimal shifts) in the backbone residues to ensure we can link neighboring residues. We agree that future iterations of qFit should include more effective backbone sampling by exploring motion along the Cβ-Cα, C-N, and (Cβ-Cα × C-N) bonds and exploring correlated backbone movements.

(2) While the paper is well split in clear parts, some of them seem to be not at their right/optimal place and better can be moved to "Methods" (detailed "Overview of the qFit protein algorithm" as a whole) or to "Data" missed now (Two first paragraphs of "qFit improves overall fit...", page 8, and "Generating the qFit test set", page 22, and "Generating synthetic data ..." at page 26; description of the test data set), At my personal taste, description of tests with simulated data (page 15) would be better before that of tests with real data.

Thank you for this comment, but we stand by our original decision to keep the general flow of the paper as it was submitted.

(3) I wonder if the term "quadratic programming" (e.g., A3, page 5) is appropriate. It supposes optimization of a quadratic function of the independent parameters and not of "some" parameters. This is like the crystallographic LS which is not a quadratic function of atomic coordinates, and I think this is a similar case here. Whatever the answer on this remark is, an example of the function and its parameters is certainly missed.

We think that the term quadratic programming is appropriate. We fit a function with a loss function (observed density - calculated density), while satisfying the independent parameters. We fit the coefficients minimizing a quadratic loss. We agree that the quadratic function is missing from the paper, and we have now included it in the Methods section.

Technical remarks to be answered by the authors :(1) Page 1, Abstract, line 3. The ensemble modeling is not the only existing frontier, and saying "one of the frontiers" may be better. Also, this phrase gives a confusing impression that the authors aim to predict the ensemble models while they do it with experimental data.

We agree with this statement and have re-worded the abstract to reflect this.

(2) Page 2. Burling & Brunger (1994) should be cited as predecessors. On the contrary, an excellent paper by Pearce & Gros (2021) is not relevant here.

While we agree that we should mention the Burling & Brunger paper and the Pearce & Gros (2021) should not be removed as it is not discussing the method of ensemble refinement.

(3) Page 2, bottom. "Further, when compared to ..." The preference to such approach sounds too much affirmative.

We have amended this sentence to state:

“Multiconformer models are notably easier to modify and more interpretable in software like Coot(Emsley et al. 2010) unlike ensemble methods that generate multiple complete protein copies(Burnley et al. 2012; Ploscariu et al. 2021; Temple Burling and Brünger 1994).”

“The point we were trying to make in this sentence was that ensemble-based models are much harder to manually manipulate in Coot or other similar software compared to multiconformer models. We think that the new version of this sentence states this point more clearly.”

(4) Page 2, last paragraph. I do not see an obvious relation of references 15-17 to the phrase they are associated with.

We disagree with this statement, and think that these references are appropriate.

“Multiconformer models are notably easier to modify and more interpretable in software like Coot12 unlike ensemble methods that generate multiple complete protein copies (Burnley et al. 2012; Ploscariu et al. 2021; Temple Burling and Brünger 1994).”

(5) Page 3, paragraph 2. Cryo-EM maps should be also "high-resolution"; it does not read like this from the phrase.

We agree that high-resolution should be added, and the sentence now states:

“However, many factors make manually creating multiconformer models difficult and time-consuming. Interpreting weak density is complicated by noise arising from many sources, including crystal imperfections, radiation damage, and poor modeling in X-ray crystallography, and errors in particle alignment and classification, poor modeling of beam induced motion, and imperfect detector Detector Quantum Efficiency (DQE) in high-resolution cryo-EM.”

(6) Page 3, last paragraph before "results". The words "... in both individual cases and large structural bioinformatic projects" do not have much meaning, except introducing a self-reference. Also, repeating "better than 2 A" looks not necessary.

We agree that this was unnecessary and have simplified the last sentence to state:

“With the improvements in model quality outlined here, qFit can now be increasingly used for finalizing high-resolution models to derive ensemble-function insights.”

(7) Page 3. "Results". Could "experimental" be replaced by a synonym, like "trial", to avoid confusing with the meaning "using experimental data"?

We have replaced experimental with exploratory to describe the use of qFit on CryoEM data. The statement now reads:

“For cryo-EM modeling applications, equivalent metrics of map and model quality are still developing, rendering the use of qFit for cryo-EM more exploratory.”

(8) Page 4, A.1. Should it be "steps +/- 0.1" and "coordinate" be "coordinate axis"? One can modify coordinates and not shift them. I do not understand how, with the given steps, the authors calculated the number of combinations ("from 9 to 81"). Could a long "Alternatively, ...absent" be reduced simply to "Otherwise"?

We have simplified and clarified the sentence on the sampling of backbone coordinates to state:

“If anisotropic B-factors are absent, the translation of coordinates occurs in the X, Y, and Z directions. Each translation takes place in steps of 0.1 along each coordinate axis, extending to 0.3 Å, resulting in 9 (if isotropic) or to 81 (if anisotropic) distinct backbone conformations for further analysis.”

(9) Page 6, B.1, line 2. Word "linearly" is meaningless here.

We have modified this to read:

“Moving from N- to C- terminus along the protein,”

(10) Page 9, line 2. It should be explained which data set is considered as the test set to calculate Rfree.

We think this is clear and would be repetitive if we duplicated it.

(11) Page 9, line 7. It should be "a valuable metric" and not "an"

We agree and have updated the sentence to read:

“Rfree is a valuable metric for monitoring overfitting, which is an important concern when increasing model parameters as is done in multiconformer modeling.”

(12) Page 10, paragraph 3. "... as a string (Methods)". I did not find any other mention of this term "string", including in "Methods" where it supposed to be explained. Either this should be explained (and an example is given?), or be avoided.

We agree that string is not necessary (discussing the programmatic datatype). We have removed this from the sentence. It now reads:

“To quantify how often qFit models new rotameric states, we analyzed the qFit models with *phenix.rotalyze*, which outputs the rotamer state for each conformer (**Methods**).”

(13) Page10, lines 3-4 from bottom. Are these two alternative conformations justified?

We are unsure what this is referring to.

(14) Page 12, Fig. 2A. In comparison with Supplement Fig 2C, the direction of axes is changed. Could they be similar in both Figures?

We have updated Supplementary Figure 2C to have the same direction of axes as Figure 2A.

(15) Page 15, section's title. Choose a single verb in "demonstrate indicate".

We have amended the title of this section to be:

“Simulated data demonstrate qFit is appropriate for high-resolution data.”

(16) Page 15, paragraph 2. "Structure factors from 0.8 to 3.0 A resolution" does not mean what the author wanted apparently to tell: "(complete?) data sets with the high-resolution limit which varied from 0.8 to 3.0 A ...". Also, a phrase of "random noise increasing" is not illustrated by Figs.5 as it is referred to.

We have edited this sentence to now read:

“To create the dataset for resolution dependence, we used the ground truth 7KR0 model, including all alternative conformations, and generated artificial structure factors with a high resolution limit ranging from 0.8 to 3.0 Å resolution (in increments of 0.1 Å).”

(17) Page 15, last paragraph is written in a rather formal and confusing way while a clearer description is given in the figure legend and repeated once more in Methods. I would suggest to remove this paragraph.

We agree that this is confusing. Instead of create a true positive/false positive/true negative/false negative matrix, we have just called things as they are, multiconformer or single conformer and match or no match. We have edited the language the in the manuscript and figure legends to reflect these changes.

(18) Page 16. Last two paragraphs start talking about a new story and it would help to separate them somehow from the previous ones (sub-title?).

We agree that this could use a subtitle. We have included the following subtitle above this section:

“Simulated multiconformer data illustrate the convergence of qFit.”

(19) Page 20. "or static" and "we determined that" seem to be not necessary.

We have removed static and only used single conformer models. However, as one of the main conclusions of this paper is determining that qFit can pick up on alternative conformers that were modeled manually, we have decided to the keep the “we determined that”.

(20) Page 21, first paragraph. "Data" are plural; it should be "show" and "require"

We have made these edits. The sentence now reads:

“However, our data here shows that not only does qFit need a high-resolution map to be able to detect signal from noise, it also requires a very well-modeled structure as input.”

(21) Page 21, References should be indicated as [41-45], [35,46-48], [55-57]. A similar remark to [58-63] at page 22.

We have fixed the reference layout to reflect this change.

(22) Page 21, last paragraph. "Further reduce R-factors" (moreover repeated twice) is not correct neither by "further", since here it is rather marginal, nor as a goal; the variations of R-factors are not much significant. A more general statement like "improving fit to experimental data" (keeping in mind density maps) may be safer.

We agree with the duplicative nature of these statements. We have amended the sentence to now read:

“Automated detection and refinement of partial-occupancy waters should help improve fit to experimental data further reduce Rfree15 and provide additional insights into hydrogen-bond patterns and the influence of solvent on alternative conformations.”

(23) Page 22. Sub-sections of "Methods" are given in a little bit random order; "Parallelization of large maps" in the middle of the text is an example. Put them in a better order may help.

We have moved some section of the Methods around and made better headings by using an underscore to highlight the subsections (*Generating and running the qFit test set, qFit improved features, Analysis metrics, Generating synthetic data for resolution dependence).*

(24) Page 24. Non-convex solution is a strange term. There exist non-convex problems and functions and not solutions.

We agree and we have changed the language to reflect that we present the algorithm with non-convex problems which it cannot solve.

(25) Page 26, "Metrics". It is worthy to describe explicitly the metrics and not (only) the references to the scripts.

For all metrics, we describe a sentence or two on what each metric describes. As these metrics are well known in the structural biology field, we do not feel that we need to elaborate on them more.

(26) Page 26. Multiplying B by occupancy does not have much sense. A better option would be to refer to the density value in the atomic center as occ*(4*pi/B)^1.5 which gives a relation between these two entities.

We agree and have update the B-factor figures and metrics to reflect this.

(27) Page 40, suppl. Fig. 5. Due to the color choice, it is difficult to distinguish the green and blue curves in the diagram.

We have amended this with the colors of the curves have been switched.

(28) Page 42, Suppl. Fig. 7. (A) How the width of shaded regions is defined? (B) What the blue regions stand for? Input Rfree range goes up to 0.26 and not to 0.25; there is a point at the right bound. (C) Bounds for the "orange" occupancy are inversed in the legend.

(A) The width of the shaded region denotes the standard deviations among the values at every resolution. We have made this clearer in the caption

(B) The blue region denotes the confidence interval for the regression estimate. Size of the confidence interval was set to 95%. We have made this clearer in the caption

(C) This has been fixed now

The maximum R-free value is 0.2543, which we rounded down to 0.25.

(29) Page 43. Letters E-H in the legend are erroneously substituted by B-E.

We apologize for this mistake. It is now corrected.